# Ribosomal RNA synthesis by RNA polymerase I is subject to premature termination of transcription

**Chaïma Azouzi[1], Katrin Schwank[2], Sophie Queille[1], Marta Kwapisz[1], Marion Aguirrebengoa[3], Anthony Henras[2], Simon Lebaron[1], Herbert Tschochner[2], Annick Lesne[4], Frederic Beckouët[1], Olivier Gadal[1]\*, Christophe Dez[1]\***

[1]University of Toulouse, CNRS, Centre de Biologie Integrative (CBI), Toulouse, France; [2]Universität Regensburg, Regensburg Center of Biochemistry (RCB), Lehrstuhl Biochemie III, Regensburg, Germany; [3]BigA, University of Toulouse, CNRS, Centre de Biologie Integrative (CBI), Toulouse, France; [4]Laboratoire de Physique Théorique de la Matière Condensée (LPTMC), Sorbonne Université, Paris, France

## eLife Assessment

This manuscript characterizes a mutated clone of RNA polymerase I in yeast, referred to as SuperPol, to understand the mechanisms of RNA polymerase I elongation and termination. The authors present **convincing** evidence that demonstrates the existence of premature termination in Pol I transcription. Overall, the characterization of this RNA pol I offers **important** insights into the regulation of ribosomal RNA transcription and its potential application in cancer pharmacology.
[Editors' note: this paper was reviewed by Review Commons.]

**\*For correspondence:**
olivier.gadal@univ-tlse3.fr (OG);
christophe.dez@univ-tlse3.fr
(CD)

**Competing interest:** The authors declare that no competing interests exist.

**Abstract** The RNA polymerase I (Pol I) enzyme that synthesizes large rRNA precursors exhibits a high rate of pauses during elongation, indicative of a discontinuous process. We show here that premature termination of transcription (PTT) by Pol I in yeast *Saccharomyces cerevisiae* is a critical regulatory step limiting rRNA production in vivo. The Pol I mutant, SuperPol (RPA135-F301S), produces 1.5-fold more rRNA than the wild type (WT). Combined CRAC and rRNA analysis link increased rRNA production in SuperPol to reduced PTT, resulting in shifting polymerase distribution toward the 3' end of rDNA genes. In vitro, SuperPol shows reduced nascent transcript cleavage, associated with more efficient transcript elongation after pauses, to the detriment of transcriptional fidelity. Notably, SuperPol is resistant to BMH-21, a drug impairing Pol I elongation and inducing proteasome-mediated degradation of Pol I subunits. Compared to WT, SuperPol maintains subunit stability and sustains high transcription levels upon BMH-21 treatment. These comparative results show that PTT is alleviated in SuperPol while it is stimulated by BMH-21 in WT Pol I.

## Introduction

Ribosome biogenesis is a major energy-consuming process in exponentially growing cells. In eukaryotes, making a ribosome requires the coordinated activities of the three eukaryotic nuclear RNA polymerases, accounting for up to 75% of total cellular transcriptional activity (*Warner, 1999*). Half of all RNA polymerase II (Pol II) initiation events are devoted to the production of mRNAs encoding ribosomal proteins and ribosome biogenesis factors. Furthermore, a prodigious amount of ribosomal RNAs (rRNAs) is synthesized by the combined transcriptional activity of the RNA polymerase I (Pol I), which leads to the production of the three large rRNAs, namely 18S, 5.8S, and 25S rRNAs, as well as

the RNA polymerase III (Pol III), which synthesizes the 5S rRNA. In the yeast *Saccharomyces cerevisiae*, the first step of the ribosome assembly process is the production of 35S rRNA (corresponding to 47S in human) by Pol I. Up to hundreds of Pol I enzymes can simultaneously transcribe a single rDNA gene. During Pol I transcription, nascent 35S pre-rRNA is folded and assembled with ribosomal proteins into large pre-ribosomes by several ribosome biogenesis factors. Within pre-ribosomes, rRNAs are co-transcriptionally matured by endonucleolytic cleavages and exonucleolytic trimming (*Henras et al., 2015*; *Kos and Tollervey, 2010*; *Woolford and Baserga, 2013*). A huge amount of cell resources is invested in ribosome production, making its regulation crucial for cell homeostasis. Importantly, the production and assembly of nearly 80 ribosomal proteins with four different rRNAs imposes regulatory challenges that must be adapted to fluctuating environmental conditions. To cope with such massive production, Pol I activity should be highly regulated. Miller chromatin spreads, allowing single molecule analysis of the rDNA gene transcribed by Pol I, clearly demonstrate the exceptional efficiency of transcription initiation by Pol I in ideal growth conditions (*Albert et al., 2011*). In fact, such initiation level leads to almost 50% occupancy of Pol I along the length of the rDNA (*French et al., 2003*).

In both yeast and mammals, essential factors required for Pol I initiation were identified through genetic and biochemical approaches (*Grummt, 2003*; *Moss et al., 2007*). The structures of Pol I initiation complex (pre-initiation complex and initially transcribing complex) are now available (*Engel et al., 2018*). Although Pol I transcription is well known to be regulatable at the initiation level (*Blattner et al., 2011*; *Sadian et al., 2019*; *Schneider et al., 2007*), rRNA synthesis and pre-rRNA processing are, at least partially, controlled during the elongation phase of Pol I transcription (*Azouzi et al., 2021*; *Schneider et al., 2007*). A statistical view of Pol I pausing within rDNA was obtained using high-throughput methods such as Net-Seq or CRAC (*Clarke et al., 2018*; *Turowski et al., 2020*). Similar to other RNA polymerases, the Pol I elongation process is based on Brownian ratchet in which energy-driven mechanism favors forward rather than backward steps. As reported by several studies, Pol I is prone to frequent pausing events and/or regions with slower elongation rates possibly accompanied by backtracking (*Clarke et al., 2018*; *Dangkulwanich et al., 2013*; *Duval et al., 2023*; *Guajardo and Sousa, 1997*; *Scull et al., 2020*; *Turowski et al., 2020*). The dynamic of Pol I elongation is influenced by various factors, such as the folding of the nascent rRNA, as well as the supercoiling of the rDNA template, as shown by biophysical modeling recapitulating Pol I density along the gene (*Kim et al., 2019*; *Lesne et al., 2018*; *Liu and Wang, 1987*; *Turowski et al., 2020*). This pausing pattern results in a strong accumulation of Pol I elongation complex in the 5' end of the rDNA gene. The high density of polymerases in the 5' ETS region is also correlated with major early pre-rRNA assembly events occurring in the 5' region of nascent rRNA (*Chaker-Margot et al., 2017*). Another area of high Pol I density on the rDNA can be observed between the 18S and 25S rRNA genes. These areas of high pausing frequency also correspond to a crucial co-transcriptional assembly event, leading to the completion of the small ribosomal subunit rRNA precursor (20S pre-rRNA), and the production of the rRNA molecules composing the large ribosomal subunit (5.8S and 25S). Such pausing patterns correlate with major pre-ribosome assembly steps and also suggest a high level of optimization of Pol I activity.

Using genetic tools, we could show that wild-type (WT) Pol I transcriptional activity, which is the highest among the three nuclear RNA polymerases, can be further increased. We recently identified a Pol I mutant bearing a single amino acid substitution on the second largest subunit: *RPA135-F301S* allele (*Darrière et al., 2019*), hereafter named SuperPol. This mutant was isolated as an extragenic suppressor of growth defect of Pol I mutant lacking subunits Rpa49 (orthologs of human PAF53). This mutation increases rRNA production in yeast, measured by transcriptional run-on. Surprisingly, using either Miller spread or ChIP techniques, we were unable to detect any significant variation in the loading rate of Pol I on rDNA in the mutant compared to the WT. Importantly, an accumulation of 35S rRNA in cells bearing SuperPol was only observed in the absence of Rrp6, the nuclear component of the exosome that is involved in rRNA decay (*Darrière et al., 2019*).

In this study, we first set up a technique we called Pol I transcriptional monitoring assay (TMA) to confirm a 50% increase of rRNA synthesis in SuperPol relative to WT background in living cells. We used the CRAC method to demonstrate that the SuperPol mutant is prone to fewer stalling events in the 5' end of the gene, compared to a WT Pol I. This modified pausing pattern correlates with a decreased premature termination of the transcription (PTT) for SuperPol relative to WT Pol I. This decreased PTT results in a global shift of the elongation complexes toward the 3' end of the gene. This suggests an increased processivity of the mutant, allowing for greater production of full-length

35S rRNA compared to the WT. Using purified WT Pol I and SuperPol in vitro, we observed a modified cleavage pattern on artificial templates. This modified activity is associated with more efficient transcript elongation after pausing, at the expense of transcriptional fidelity for SuperPol compared to WT Pol I.

Finally, we demonstrated that SuperPol is resistant to the drug BMH-21, which is known to be a poison for Pol I elongation (*Colis et al., 2014*; *Jacobs et al., 2022*; *Peltonen et al., 2014a*; *Peltonen et al., 2014b*; *Wei et al., 2018*). We could show that BMH-21 strongly promotes Pol I PTT. Altogether, these results suggest that Pol I can be regulated by PTT, and that BMH-21 targets stalled/paused Pol I and stimulates their premature termination in vivo.

## Results

### Cells bearing SuperPol overproduce rRNA in vivo

We previously suggested that SuperPol is overproducing rRNA compared to WT in vitro (*Darrière et al., 2019*). This was documented using transcription run-on (TRO) that showed an increased transcriptional activity of SuperPol over WT enzyme. In the TRO experiment, N-lauroylsarcosine is used to permeabilize cells but also induces RNAases inhibition. Surprisingly, using northern blot and short in vivo labeling experiments, we could not reveal increased pre-rRNA and/or mature rRNA accumulation in vivo due to SuperPol. We thus hypothesized that in vivo overproduced rRNAs in cells bearing SuperPol are rapidly targeted by the nuclear exosome or other cellular exonucleases. This hypothesis was supported by the detection of increased pre-rRNA production in SuperPol over WT Pol I in the absence of Rrp6, the nuclear component of the exosome. rRNAs in synthesis still attached to the polymerase, as well as abortive and partially degraded/processed transcripts, result in a faint and smeared signal corresponding to heterogeneous RNA lengths distributed along the electrophoresis gel. Due to the short half-life of rRNAs precursor, these heterogeneous RNAs represent a significant part of Pol I transcriptional activity. In order to detect such RNAs produced by Pol I and to definitely confirm our hypothesis, we set up an experiment we called Pol I TMA. This assay is initiated with a short in vivo pulse labeling of all newly synthesized RNAs using incorporation of Phosphorus-32 ($[^{32}P]$). Neosynthesized, radiolabeled RNAs are next extracted, partially fragmented, and used to probe slot-blots loaded with single-stranded DNA fragments complementary to rDNA locus. *Figure 1A* shows the position of the different probes along rDNA. The use of slot-blots like in TRO, instead of gel electrophoresis to visualize radiolabeled RNAs, allowed the monitoring of the heterogeneous rRNA transcripts irrespective of their synthesis/degradation status, in addition to pre-rRNAs and mature rRNAs (see *Figure 1*). In contrast to TRO, neo-synthesized transcripts can be degraded following incorporation of radiolabeled nucleotide. To minimize the effect of degradation in TMA readout, we determine the shortest time window required to detect labeled rRNA between $[^{32}P]$ pulse and extraction of nascent RNA. A slight radioactive signal was already detectable after a 20 s labeling, but we chose to pulse for 40 s to get a significant and reproducible signal.

Pol I TMA was performed several times using WT and cells bearing SuperPol. We used 5S rRNA signal as internal control for normalization in all further experiments. As previous analysis indicated that incorporation of $[^{32}P]$ in the 5S rRNA transcribed by RNA Pol III was mildly detectable (*Briand et al., 2001*) (data not shown, also observed using TRO; *Wery et al., 2009*), we used two probes covering the entire 5S rRNA sequence to get decent signal. In parallel, the IGS2 signal was used as a negative control.

This novel method allowed us to confirm our in vivo initial TRO measurement: the observation of an increased labeled rRNA signal for Pol I transcript in Superpol, even in the presence of Rrp6. Results from three independent experiments were quantified and analyses revealed a 50% increase of Pol I transcription in vivo, in cells bearing SuperPol compared to WT control (*Figure 1B and C*). Note that this value is probably underestimated as degradation of some of the different transcripts still occurs during the 40 s pulse. Overall, these results show that the F301S mutation in Rpa135 drastically increases Pol I capability to produce rRNA. Pol I TMA is clearly an efficient method to evaluate transcriptional activity in vivo.

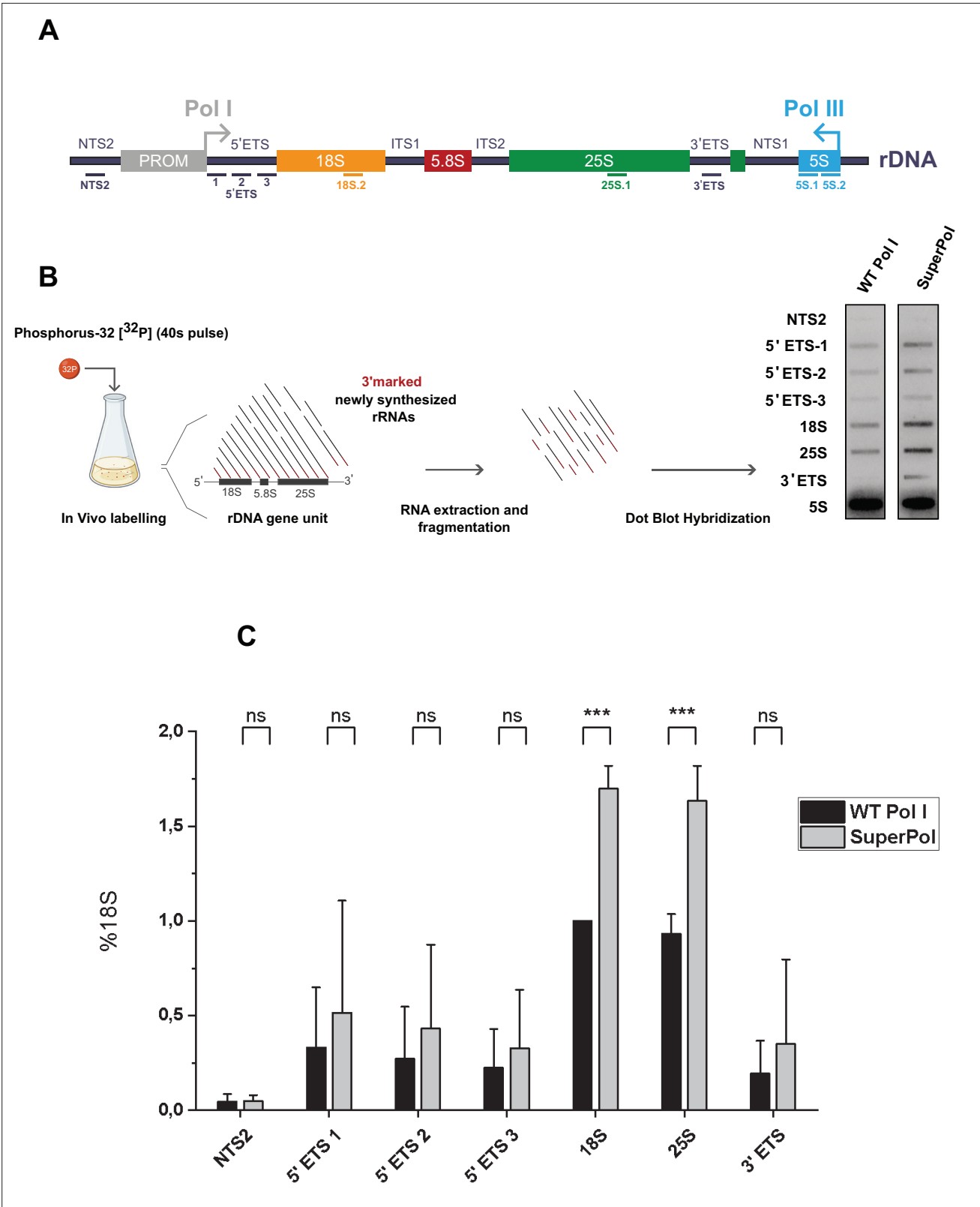

**Figure 1.** Cells bearing SuperPol overproduce ribosomal RNA (rRNA) in vivo. (**A**) Yeast rDNA unit is represented, with the position of the corresponding antisense oligonucleotides used to load slot-blots (see Materials and methods for description). (**B**) Pol I transcriptional monitoring assay (TMA) was performed in RPA135:F301S and otherwise isogenic wild-type cells grown to mid-log phase in phosphate-depleted YPD medium. Nascent transcripts were labeled with Phosphorus-32 ([32P]) for 40 s. 3' marked newly synthesized RNAs were extracted, partially hydrolyzed, and revealed using slot-blots.

*Figure 1 continued on next page*

*Figure 1 continued*

Each labeling was performed independently in triplicate; one representative example is shown in the panel. (**C**) Results represented in panel B were quantitated. IGS2 and Pol I (5'ETS, 18S.2, 25S.1, and 3' ETS) are quantified relative to 5S signal in the lower panel. Error bars indicate mean ± SD. ***p<0.005; **p<0.01; *p<0.05, calculated by two-sample t-test.

The online version of this article includes the following source data for figure 1:

**Source data 1.** Original file containing slot Blots for *Figure 1B*.

**Source data 2.** Original file containing slot Blots for *Figure 1B*, indicating the relevant lanes used.

## SuperPol displays a lower rate of pause compared to WT Pol I

As demonstrated previously by Miller's spread experiment and ChIP, the number of polymerases engaged in transcription in strains bearing WT Pol I and SuperPol is similar (*Darrière et al., 2019*). An increased rRNA production with a comparable number of Pol I engaged in transcription suggests a modification of elongation properties of SuperPol relative to WT. Thus, we compared WT Pol I and SuperPol distribution during elongation using the CRAC method developed by *Granneman et al., 2009*, and adapted to Pol I by *Turowski et al., 2020*. This technique reveals Pol I distribution at nucleotide level along the rDNA gene. Assuming a steady state, the velocity can be interpreted from the spatial distribution. Thus, it allows to visualizing the propensity of the polymerase to pause or slow down during elongation.

The WT Pol I distribution profile we obtained is qualitatively highly similar to the one published by *Turowski et al., 2020*, showing the robustness of the CRAC analysis (*Figure 2A*, *Figure 2—figure supplement 1*). When comparing WT Pol I and SuperPol global spatial distribution profiles along rDNA gene, massive accumulations are visible in 5'ETS, and to a lower extent between the 5.8S sequence and the beginning of the 25S sequence. Zooming on the 5' end of the rDNA gene, we could observe SuperPol and WT Pol I pausing at identical positions, but not to the same extent (*Figure 2B*, *Figure 2—figure supplement 1*). To compare the most frequent pausing sites of Pol I and SuperPol, we computed the differences between their CRAC profiles (gray curve). This analysis revealed that SuperPol exhibits reduced accumulation at various positions between the transcription start site (TSS) and 300th nucleotide. This indicates that SuperPol likely transitions more efficiently through this region, possibly due to decreased pausing or enhanced elongation efficiency during the early stages of transcription. To highlight regions of comparatively lower pausing frequency, we also computed the ratio between distributions (excluding 27 nt out of the 225 nt shown here, due to low coverage). Despite a highly reproducible global profile, we could identify local variations between Pol I CRAC nucleotide distribution replicates. Importantly, despite some local variations, we could reproducibly observe a 25% increased occupancy of WT Pol I in 5'ETS compared to SuperPol (*Figure 2C* and *Figure 2—figure supplement 1*). Overall, analysis of CRAC data did not allow identifying specific sequence feature at which SuperPol and WT were differentially accumulated. However, the analysis of occupancy of SuperPol relative to WT Pol I revealed a significant decrease in the 5'ETS.

## Modified elongation properties of SuperPol impact PTT

This analysis of Pol I distribution using CRAC showed that increased rRNA synthesis of SuperPol is associated with modified elongation dynamics. The concept of PTT as a transcriptional control of gene expression has long been established in bacteria and is referred to as 'attenuation' (*Artz and Broach, 1975*; *Bertrand et al., 1975*). In yeast and mammalian cells, the study of the transcription by RNA Pol II and III revealed PTT as a major contributor to global transcriptional regulation (*Arigo et al., 2006*; *El Hage et al., 2008*; *Kamieniarz-Gdula and Proudfoot, 2019*; *Porrua and Libri, 2015*; *Steinmetz et al., 2006*; *Xie et al., 2023*).

The occurrence of PTT during RNA Pol I transcription has only been reported following drugs that inhibit Pol I elongation, such as actinomycin D (*Fetherston et al., 1984*; *Shcherbik et al., 2010*). However, WT Pol I and SuperPol exhibit similar transcription initiation properties (*Darrière et al., 2019*) but different pause kinetics (*Figure 2B*). We hypothesized that the overproduction of rRNA in strains expressing SuperPol may stem from a reduced frequency of PTT occurrences due to shorter pauses.

Since abortive transcripts are released from transcribing Pol I, they cannot be detected by the CRAC Pol I method. Therefore, to identify abortive transcripts, we performed northern blot analysis.

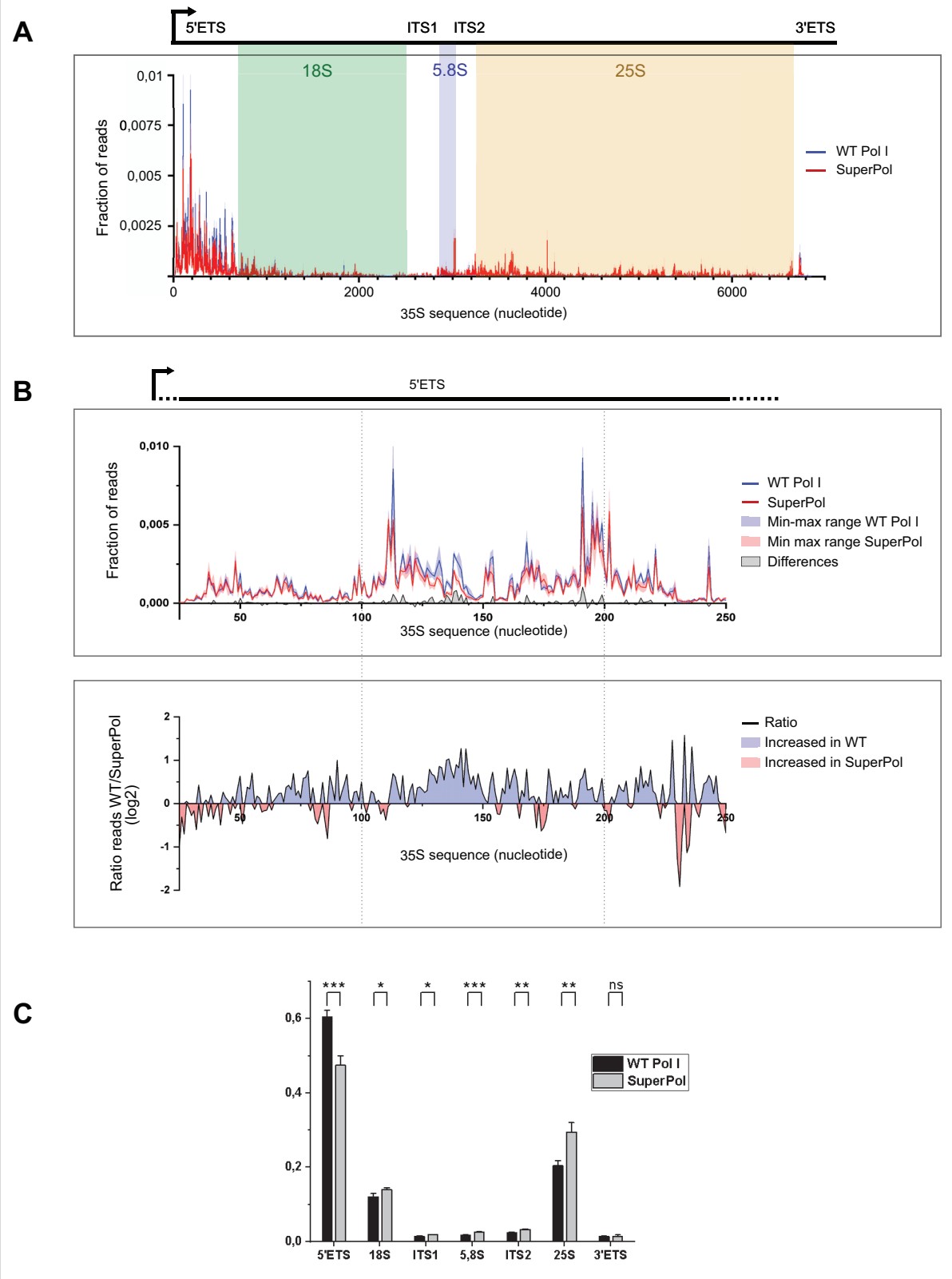

**Figure 2.** SuperPol displays a lower rate of pause compared to wild-type (WT) polymerase I (Pol I). (**A**) CRAC distribution profiles obtained for WT Pol I (blue line) and SuperPol (red line). Lines correspond to the mean frequency of reads obtained for three independent experiments. The area of the 35S rDNA gene has been highlighted (5'ETS from +1 to 700 nt; 18S from 701 to 2499 nt; ITS1 from 2500 to 2858 nt; 5.8S from 2859 to 3019 nt; ITS2 from 3020 to 3250 nt; 25S from 3251 to 6647 nt; 3'ETS from 6648 to 6859 nt). (**B**) Zoom-in 5'ETS CRAC distribution profiles obtained for WT Pol I (blue) and

*Figure 2 continued on next page*

*Figure 2 continued*

SuperPol (red). Lines correspond to the mean frequency of reads obtained for three independent experiments, while the colored area corresponds to the min-max range of the three experiments. The gray area corresponds to the difference between WT Pol I and SuperPol frequency of reads at each position of the gene. This difference has been calculated using the min-max range values, giving the lowest difference (white area between blue and red min-max range). The lower panel represents the ratio between WT Pol I and SuperPol, represented in log2. Blue areas correspond to sequences where WT is more accumulated than SuperPol and red areas correspond to the sequences where SuperPol is more accumulated than WT Pol I. (C) Frequency of reads obtained for each 35S area (5'ETS, 18S, ITS1, 5.8S, ITS2, 25S, 3'ETS) from three independent experiments was summed and shown as mean ± SD. ***p<0.005; **p<0.01; *p<0.05, calculated by two-sample t-test.

The online version of this article includes the following source data and figure supplement(s) for figure 2:

**Source data 1.** Excel spreadsheet containing the quantitative analysis of the CRAC experiment shown in *Figure 2*.

**Figure supplement 1.** SuperPol displays a lower rate of pause compared to wild-type (WT) polymerase I (Pol I).

**Figure supplement 1—source data 1.** Excel spreadsheet containing the quantitative analysis of the CRAC experiment shown in *Figure 2—figure supplement 1*.

PTT is known to produce abortive transcripts that are generally unprocessed. Once released from Pol I, their 3' ends become accessible to cellular exonucleases. As a result, these RNAs have a very short half-life and only accumulate when 3'–5' exonucleases, such as the RNA exosome, are inhibited (*Kamieniarz-Gdula and Proudfoot, 2019*, and see *Figure 3A*). We therefore undertook a search for abortive rRNAs in order to compare their accumulation in strains expressing WT Pol I or SuperPol. According to our hypothesis, such transcripts would be produced in higher amounts in WT strains compared to mutants bearing SuperPol. To reveal abortive transcripts, we performed northern blot analysis in the presence and absence of Rrp6, the nuclear-specific 3'–5' exonuclease subunit of the RNA exosome (*Figure 3B*). Therefore, Rrp6-sensitive accumulation of rRNA indicates transcripts with an accessible 3'-end.

This experiment reveals the existence of such transcripts at the proximity of TSS. The size of one of them is comprised between 70 and 90 nt and is accumulated in absence of Rrp6 (*Figure 3B*, lane 3). Interestingly, the accumulation of these abortive transcripts is greatly reduced in SuperPol background (*Figure 3B*, lane 4, quantified in *Figure 3C*). Note that, other known rRNA species resulting from degradation products of full ETS1 by the exosome are accumulated in this area (≈130 and 160 nt long) (*Delan-Forino et al., 2017*). As opposed to abortive transcript, those degradation products are detected in both WT and SuperPol.

These results unveil a massive amount of abortive rRNAs produced by Pol I. To investigate if the altered PTT rate in SuperPol accounts for its 1.5-fold increased full-length rRNA synthesis relative to WT Pol I, a cumulative distribution function (CDF) analysis of Pol I CRAC profiles across the 6861 nt of the entire 35S rRNA (*Figure 3D*). The CDF of SuperPol shifts rightward, indicating reduced accumulation in the 5' region of the gene relative to WT Pol I. Altogether, we propose that SuperPol is more processive than WT Pol I as this low pausing rate is associated with a reduction in the quantity of abortive transcription events.

## The enzymatic properties of SuperPol result in reduced cleavage activity and increased elongation dynamics, to the detriment of transcriptional fidelity

Pol I elongation is a discontinuous and stochastic process based on Brownian ratchet motion, making it prone to frequent pausing, backtracking, and RNA cleavage to restart elongation.

In order to characterize SuperPol enzymatic properties, we first performed in vitro cleavage assays (*Merkl et al., 2020*; *Pilsl et al., 2016a*; *Pilsl et al., 2016b*; *Schwank et al., 2022*). WT Pol I and SuperPol have been purified several times from yeast cells expressing an HTP-tagged subunit Rpa135 (*Figure 4A and B* and *Figure 4—figure supplement 1*). RNA cleavage was monitored using an RNA/DNA hybrid scaffold containing 3 nt mismatch at the RNA 3'end (clv3) (*Figure 5A*). The scaffold was then incubated with purified Pol I elongation complex in the absence of ribonucleotides from 1 to 30 min. Results shown in *Figure 5B* (quantified in *Figure 5C*) show that WT Pol I cleaves off 2 nt of the RNA fragment used as substrate after 5 min. A second cleavage of 4 nt becomes predominant after 30 min. However, SuperPol cleaves mainly at the position –2, and only a few –4 RNA fragments are detected compared to WT Pol I, even after 30 min. This assay shows that SuperPol cleavage

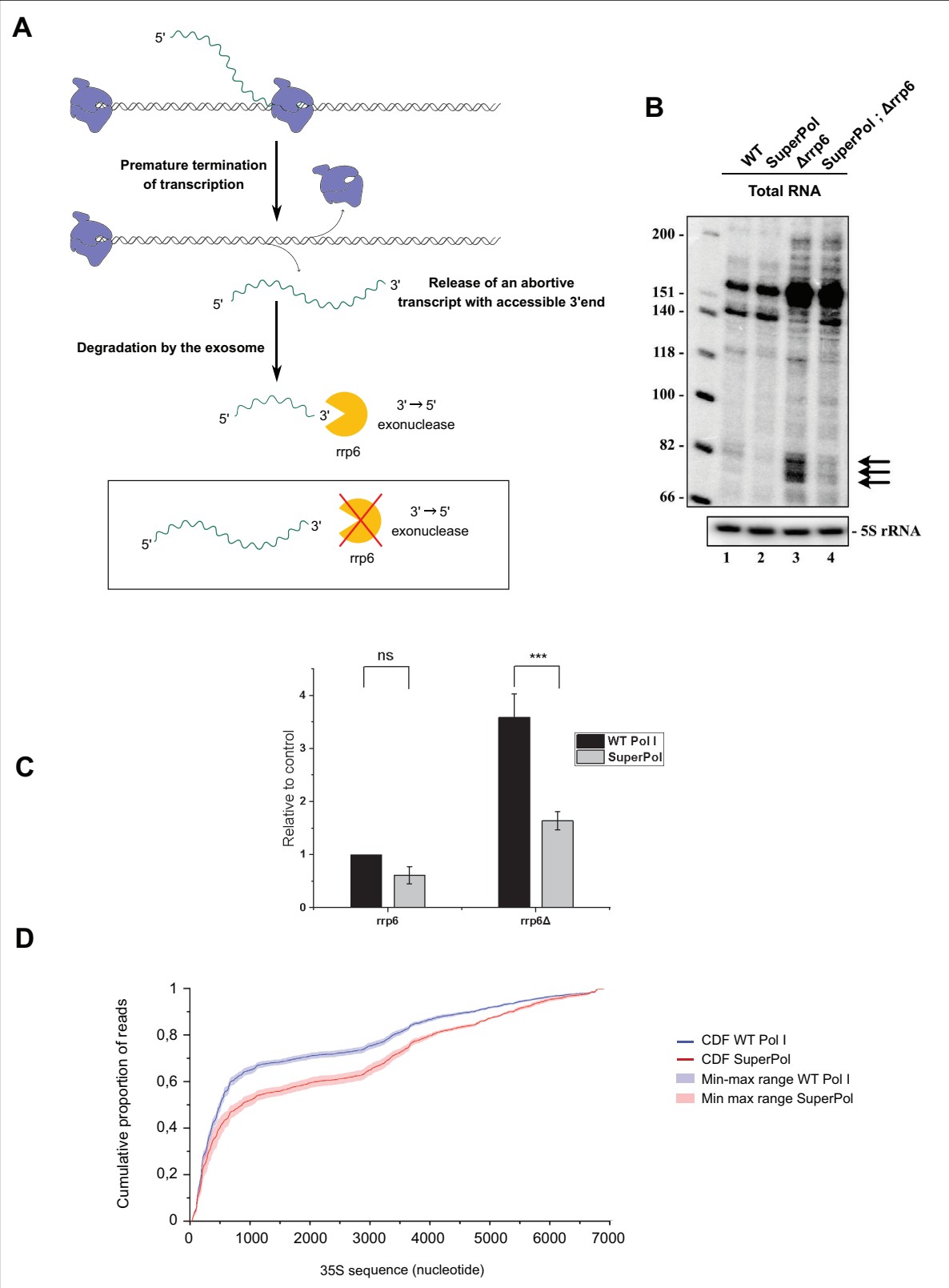

**Figure 3.** Modified elongation properties of SuperPol impact premature termination of transcription. (**A**) Events of premature terminations lead to the releasing of both stalled polymerase I (Pol I) elongation complex and nascent transcript. The 3'end of these abortive transcripts is not protected anymore by the elongation complex and becomes accessible to degradation by rrp6, the 3' to 5' exonuclease nuclear subunit of the exosome. To accumulate and detect abortive transcripts, deletion of rrp6 is necessary. (**B**) Cells bearing wild-type (WT) Pol I or SuperPol, with or without rrp6, were

*Figure 3 continued on next page*

*Figure 3 continued*

grown, and total RNAs were extracted and analyzed by northern blot using TSS probes. Low-molecular-weight RNA products were resolved on 8% polyacrylamide/8.3 M urea gels. (**C**) Low-molecular-weight RNA (≈ 80 nt) products were quantified relative to 5S signal. Error bars indicate mean ± SD. \*\*\*p<0.005; \*\*p<0.01; \*p<0.05, calculated by two-sample t-test. (**D**) Cumulative distribution function (CDF) of Pol I profiles obtained by CRAC on the 6861 nt of the rDNA. The cumulative distribution of both WT Pol I and SuperPol is represented, respectively, by the blue and red curves. Lines correspond to the mean cumulative distribution of frequency of reads obtained for three independent experiments, while the colored area corresponds to the min-max range of the three experiments.

The online version of this article includes the following source data for figure 3:

**Source data 1.** Original files containing Northern blots for *Figure 3B*.

**Source data 2.** Excel spreadsheet containing the annotated files used for *Figure 3B* and quantitative analysis of triplicates.

activity is affected in these conditions, leading to decreased ability to remove mismatched nucleotides compared to WT Pol I.

We next tested if, following cleavage, SuperPol transcriptional activity was increased using in vitro elongation assays (*Merkl et al., 2020*; *Pilsl et al., 2016a*; *Pilsl et al., 2016b*; *Schwank et al., 2023*; *Merkl et al., 2020*; *Pilsl et al., 2016a*; *Pilsl et al., 2016b*; *Schwank et al., 2022*). RNA elongation was monitored using an RNA/DNA hybrid scaffold containing 1 nt mismatch at the RNA 3'end (clv1) (*Figure 5D*). The scaffold is then incubated with purified Pol I elongation complex in the absence or presence of ribonucleotides from 1 to 30 min. The RNA, which contains a Cy5 fluorophore at its 5'end, is detected on a denaturing polyacrylamide gel.

The results show that in the absence of nucleotides, both WT Pol I and SuperPol cleave off 1 or 2 nt of the RNA fragment used as substrate after 30 min, thus removing the 3' nt mismatch (*Figure 5E*). We used this specific mismatch template because we noticed that –1 and –2 RNA fragments are detected

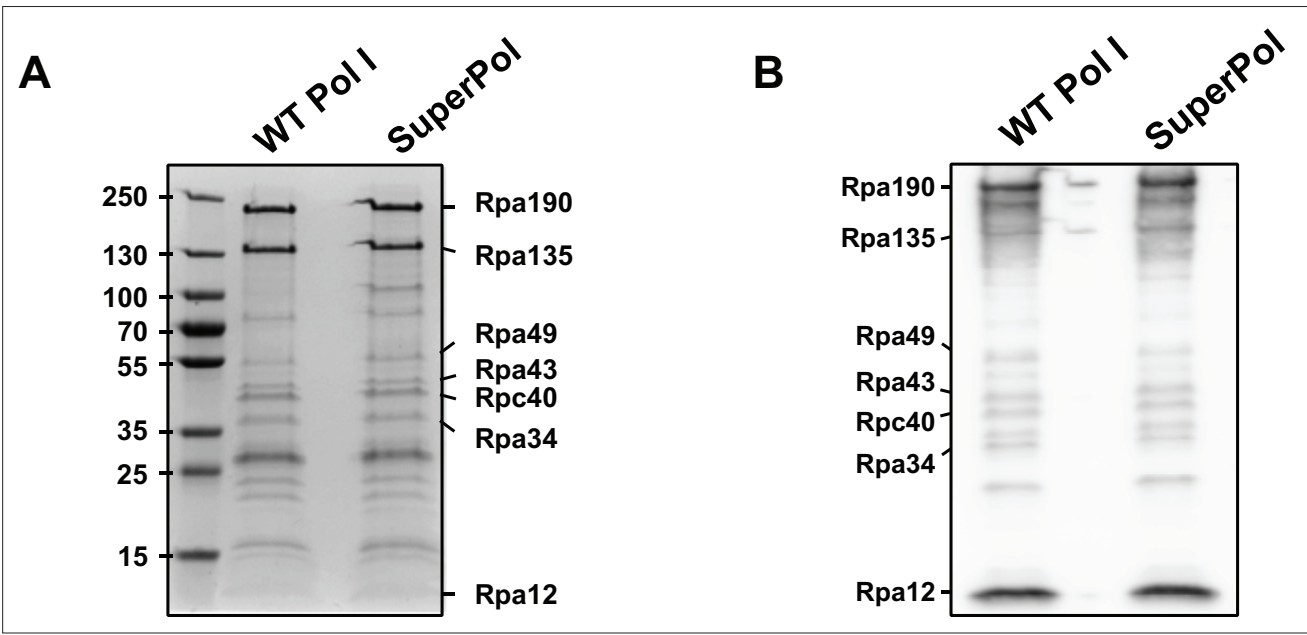

**Figure 4.** Purification of RNA polymerase I (Pol I) and SuperPol. (**A**) Tagged wild-type (WT) Pol I and SuperPol were immunopurified, respectively, from yeast strains #3207 and #3208. Purified fractions were migrated and separated on 4–12% SDS-PAGE and revealed by Coomassie Blue (see Materials and methods). (**B**) Tagged WT Pol I and SuperPol were immunopurified, respectively, from yeast strains #3207 and #3208. Purified fractions were analyzed by western blot using anti-Pol I antibody (see Materials and methods).

The online version of this article includes the following source data and figure supplement(s) for figure 4:

**Source data 1.** Original files for Coomassie and Western Blot analysis shown in *Figure 4*.

**Source data 2.** Excel spreadsheet containing the annotated files used for *Figure 4*.

**Figure supplement 1.** Purification of RNA polymerase I (Pol I) and SuperPol and in vitro elongation assay.

**Figure supplement 1—source data 1.** Original gels and blots used in *Figure 4—figure supplement 1*.

**Figure supplement 1—source data 2.** Excel spreadsheet containing the annotated files used for *Figure 4—figure supplement 1*.

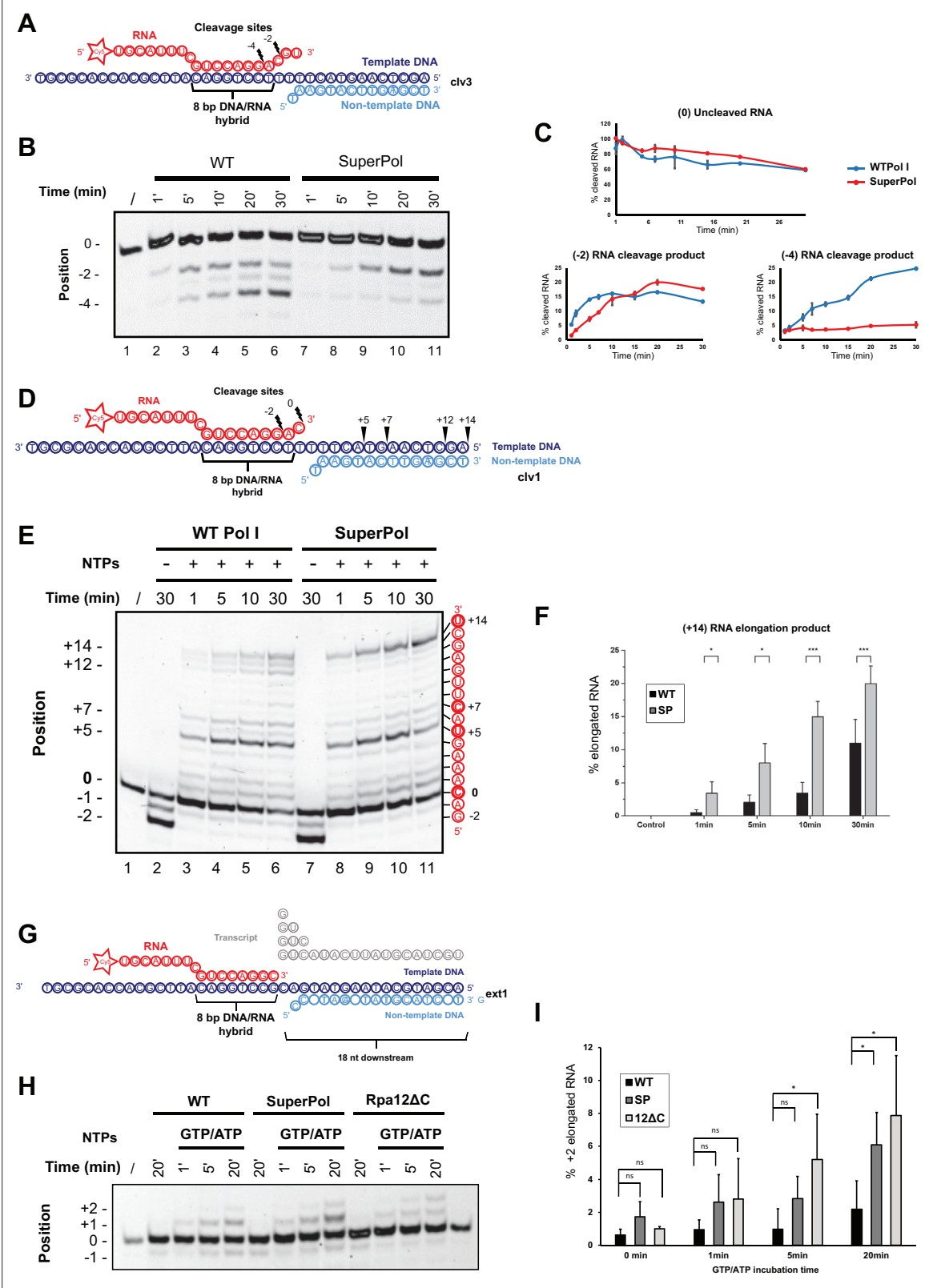

**Figure 5.** Enzymatic properties of SuperPol result in altered cleavage activity and increased elongation activity. (**A**) DNA-RNA hybrid scaffold with three mismatched nucleotides at the RNA 3'end used for cleavage assays (clv3). The RNA contains a fluorescent Cy5 label on the 5'RNA end. Cleavage sites are indicated relative to the 3'end of the RNA (site 0). (**B**) About 0.15 pmol of either wild-type (WT) polymerase I (Pol I) or SuperPol were added to 0.05 pmol cleavage scaffold and incubated for the indicated time intervals. Fluorescent transcripts were analyzed on a 20% denaturing polyacrylamide.

*Figure 5 continued on next page*

*Figure 5 continued*

RNAs shortened by two (–2) or four (–4) nucleotides and uncleaved RNAs (0) are indicated. (**C**) Quantification of cleavage efficiency (uncleaved (0) RNA) and the accumulation of –2 and –4 RNA from three independent assays. Values are indicated as a percentage of (0) uncleaved RNA after incubation of scaffold template clv3 with WT Pol I and SuperPol. (**D**) DNA-RNA hybrid scaffold with one mismatched nucleotide at the RNA 3'end used for elongation assays (clv1). The RNA contains a fluorescent Cy5 label on the 5'RNA end. Cleavage sites are indicated relative to the 3'end of the RNA (site 0). (**E**) About 0.15 pmol of either WT Pol I or SuperPol was added to 0.05 pmol cleavage scaffold and incubated with or without nucleotides (200 µM final concentration of each) for the indicated time intervals. Fluorescent transcripts were analyzed on a 20% denaturing polyacrylamide gel. Uncleaved RNAs (0), RNAs shortened by two (–2) nucleotides, and elongated RNAs (from +1 to +14) are indicated with corresponding sequence. (**F**) Quantification of elongation efficiency (+14 nt RNA elongation product) from three independent assays. Values were quantified relative to (0) uncleaved RNA after incubation of scaffold template clv1 with WT Pol I and SuperPol, with or without nucleotides. Error bars indicate mean ± SD. \***p<0.005; \*\*p<0.01; *p<0.05, calculated by two-sample t-test. (**G**) RNA-DNA hybrid scaffold including no mismatches used for misincorporation assays (ext1). The RNA contains a fluorescent Cy5 label on the 5'RNA end. Ribonucleotides to be incorporated are indicated in gray. (**H**) About 0.2 pmol Pol I WT, 0.2 pmol SuperPol, and 0.2 pmol Pol I Rpa12ΔC were incubated with 0.066 pmol RNA-DNA scaffold for 20 min on ice. Mixtures of GTP and ATP (200 pmol) were added, and samples were incubated at 28°C for the indicated time intervals. Resulting transcripts were analyzed on a 20% denaturing polyacrylamide gel. RNAs shortened by one (–1) nucleotide, uncleaved RNAs (0), and extended RNAs by one (+1) and two (+2) nucleotides are indicated. Note that production of +2 RNAs reveals misincorporation as the second nucleotide to be incorporated should be UTP and not GTP nor ATP. (**I**) Quantification of misincorporation from three independent assays. The production of elongated RNA resulting from misincorporation (+2) relative to uncleaved RNA (0) from WT, SuperPol, and Rpa12ΔC was quantified. Note that significant misincorporation of SuperPol relative to WT is detected after a 20 min incubation. Error bars indicate mean ± SD. \***p<0.005; \*\*p<0.01; *p<0.05, calculated by two-sample t-test.

The online version of this article includes the following source data for figure 5:

**Source data 1.** Original gels used in *Figure 5B, E and H*.

**Source data 2.** Excel spreadsheet containing the annotated files used for *Figure 5B, E and H*.

in the same amount with both WT Pol I and SuperPol (compare *Figure 5E*, lanes 2 and 7). When polymerases are incubated in the presence of ribonucleotides, the RNA fragment cleaved at –2 can now be used as substrate to elongate up to the end of the DNA template, resulting in the production of a +14-nt-long RNA. From 1 to 30 min of incubation with ribonucleotides, the production of this +14 nt RNA is twofold higher by the SuperPol compared to WT Pol I (*Figure 5E*, quantified in *Figure 5F*). This result shows that SuperPol displays an improved ability to resume elongation following cleavage compared to WT Pol I. This assay has been reproduced several times with the use of four independent polymerase purifications (see *Figure 4—figure supplement 1*).

Finally, we compared the error rates for WT Pol I and SuperPol during elongation. To investigate this feature in vitro, a DNA-RNA template including no mismatches (ext1; *Figure 5G*) is incubated with GTP and ATP. The GTP, which is complementary to the +1 position of the DNA template at the 3' end of the hybridized RNA, should be correctly incorporated by the elongation complex in order to produce a +1 nt RNA product. The detection of a +2 nt RNA indicates a misincorporation event. An increased +2 nt product reveals that SuperPol is more prone to misincorporate nucleotides compared to WT Pol I (*Figure 5H*). A strain lacking the C-terminal part of the subunit Rpa12, which is required for cleavage activity, was used as a control. This strain cannot cleave misincorporated nucleotides and thus displays the highest rates of misincorporation. Quantification of this assay (*Figure 5I*) allowed us to determine that SuperPol misincorporation rates are intermediate between those of WT Pol I and the WT bearing the Rpa12ΔC truncation.

Based on these in vitro transcriptional assays, we concluded that SuperPol exhibits enhanced elongation dynamics compared to WT Pol I. Specifically, the elongation assays showed that SuperPol elongates the RNA substrate at a rate approximately twofold higher than that of the WT enzyme when ribonucleotides are present. This increased elongation is sustained over time, as evidenced by the more rapid production of the +14 nt RNA fragment, reinforcing the notion that SuperPol has improved processivity both in vivo and in vitro, albeit at the expense of a higher nucleotide misincorporation than the WT enzyme.

## SuperPol is resistant to BMH-21 and retains high levels of transcription upon BMH-21 treatment

BMH-21 is a small molecule that was previously identified in a high-throughput screen for anticancer agents in Laiho's laboratory (*Peltonen et al., 2010*). It has been reported that BMH-21 is a DNA intercalator that preferentially and non-covalently binds to regions rich in GC islands, without activating

the DNA damage response (*Colis et al., 2014*; *Peltonen et al., 2010*). In mammalian cells, BMH-21 rapidly and efficiently blocks Pol I-mediated transcription and induces proteasome-dependent degradation of the largest Pol I subunit, RPA194 (*Peltonen et al., 2014b*). This effect is characteristic of BMH-21 and is conserved in the yeast *S. cerevisiae*. Treatment of a yeast strain with BMH-21 specifically affects Pol I transcription elongation, induces the degradation of Rpa190, and strongly reduces cell viability (*Wei et al., 2018*). Indeed, Pol I transcription elongation complexes appear to be a preferential substrate for Pol I subunit degradation when cells are exposed to BMH-21.

To challenge our model suggesting that SuperPol has modified elongation capabilities, we exposed mutant cells expressing SuperPol, as well as WT controls, to BMH-21 and measured cell viability, rRNA synthesis, and Pol I subunits abundance.

Cell viability was assessed using serial dilutions to evaluate number and size of individual colonies (*Figure 6A*). Growth of WT and mutant cells was indistinguishable in media without BMH-21. In the presence of a low concentration of BMH-21 (17 µM), the growth of WT cells showed a slight growth delay compared to the condition in which no BMH-21 was added. In contrast, the growth of cells bearing SuperPol was insensitive to such BMH-21 concentration. The contrast was even more pronounced when 35 µM of BMH21 was added to the growth media. While the growth of cells expressing SuperPol is only slightly affected, the fitness of WT cells was severely reduced under the same conditions. These results show that cells expressing SuperPol are resistant to BMH-21. Note that both WT and SuperPol were unable to support growth when BMH-21 was added in the media at a concentration of 50 µM.

We next performed Pol I TMA experiments to evaluate if *RPA135-F301S* mutation affects RNA Pol I activity in response to BMH-21. WT and SuperPol strains were incubated with 35 µM or 50 µM BMH-21 or DMSO for 30 min before Pol I TMA was initiated. Compared to Pol III, the Pol I activity of WT strains specifically dropped to 70% when incubated with 35 µM BMH-21 and to less than 30% in the case of treatment with 50 µM BMH-21 (*Figure 6B*). In contrast, SuperPol retained almost 80% of its activity, even after a 30 min incubation with 50 µM BMH-21.

BMH-21 exposure results in proteasome-dependent degradation of the RNA Pol I largest subunit RPA194/Rpa190 in both human and yeast cells (*Peltonen et al., 2014b*; *Wei et al., 2018*). We thus assessed Pol I subunits' accumulation in the presence of BMH-21 in *RPA135-F301S* cells and compared it to WT. The cells were treated with 35 µM of BMH-21 and aliquots were collected at 15 min, 30 min, and 2 hr. Pol I subunits' accumulation was evaluated using anti-Pol I antibody raised against the entire enzyme (*Figure 6C*). As previously observed (*Wei et al., 2018*), Rpa190 in WT cells is rapidly destabilized by proteasome-dependent degradation after 15 min of treatment with BMH-21, to reach less than 20% of its original level after 2 hr (*Figure 6D*, left panel). More surprisingly, we also observed a similar decrease in the accumulation of Rpa135 in the same condition, which led to an 80% degradation of the protein (*Figure 6D*, right panel). In contrast, the other Pol I subunits did not show such reduced accumulation. Rpa190 and Rpa135 exhibited a different behavior in cells expressing SuperPol. Although the exposure to high concentrations of BMH-21 affected the levels of these two subunits, 70% of their initial level remains. We concluded that the RPA135-F301S mutation decreased the proteasome-dependent degradation of the two largest subunits of Pol I in the presence of BMH-21, allowing a robust rRNA production in the presence of the drug.

## BMH-21 reduces Pol I occupancy through targeting of paused Pol I and stimulation of premature termination

To better understand the impact of BMH-21 on the elongation properties of WT Pol I and SuperPol, we analyzed by CRAC the enzyme distribution along rDNA and the production of abortive transcripts in the presence or absence of BMH-21 (*Figure 7*). Importantly, we used a short exposure to BMH-21 to ensure that Pol I was not massively depleted in vivo. Using Net-seq, Schneider's lab reported that BMH-21 treatment leads to a decrease in WT Pol I occupancy in vivo in rDNA spacer regions, 5' external transcribed spacer (5'ETS), internal transcribed spacer 1 (ITS1), internal transcribed spacer 2 (ITS2), 3'external transcribed spacer 2 (3'ETS) (*Jacobs et al., 2022*). CRAC results confirmed that WT Pol I occupancy has strongly decreased in these regions (*Figure 7A*). The difference between frequency of reads shows that the loss of WT Pol I accumulation in the presence of BMH-21 is spread along the 5'ETS and not located at specific positions as observed with SuperPol (see *Figure 2*). The

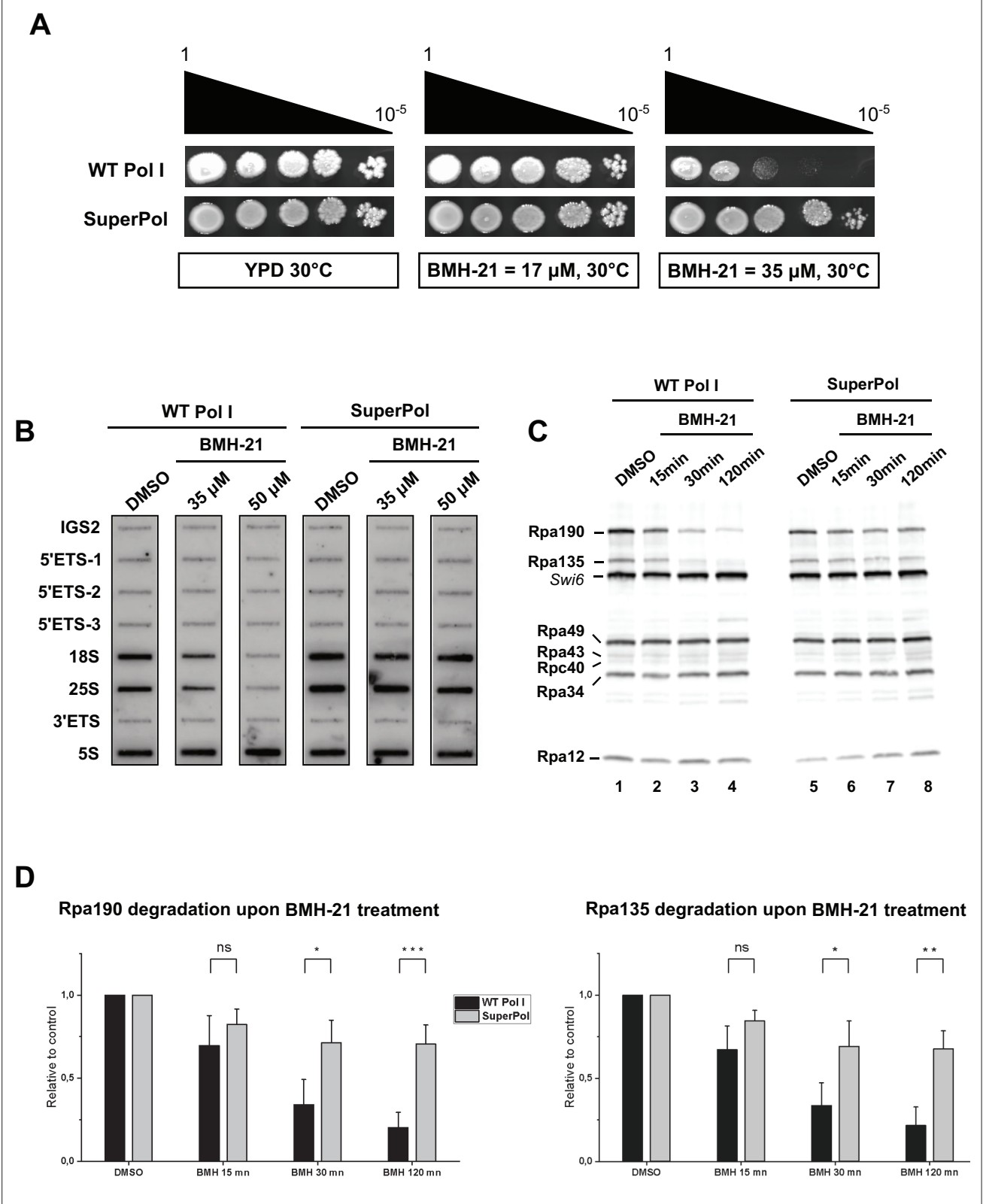

**Figure 6.** SuperPol is resistant to BMH-21 and retains high levels of transcription upon BMH-21 treatment. (**A**) SuperPol suppresses the growth defect due to BMH-21 in the yeast *S. cerevisiae*. Ten-fold serial dilutions of wild-type (WT) and RPA135-F301S single mutant were spotted on rich media to assess growth at 30°C in the presence of various concentrations of BMH-21 or vehicle. Growth was evaluated after 2 days. (**B**) Cells bearing SuperPol keep strong polymerase I (Pol I) transcriptional activity upon BMH21 treatment. Pol I transcriptional monitoring assay (TMA) allowing detection of

*Figure 6 continued on next page*

*Figure 6 continued*

accumulated newly synthesized RNA during the pulse. Nascent transcripts were labeled and detected using antisense oligonucleotides immobilized on slot-blot as described in *Figure 1* and Materials and methods. IGS2 and Pol I (mean of 5'ETS, 18S.2, 25S.1, 3' ETS) are quantified relative to 5S signal in the right panel. Yeast rDNA unit is represented in the upper panel, with the position of the corresponding antisense oligonucleotides used. (**C**) Cells bearing SuperPol maintained high levels of Rpa190 and Rpa135 upon BMH-21 treatment. WT and RPA135-F301S cells were incubated with 35 μM of BMH21 or DMSO. Aliquots were collected at 15 min, 30 min, and 2 hr, and Pol I subunits' accumulation was assessed using anti-Pol I antibody (see Materials and methods). (**D**) Accumulation of the two largest Pol I subunits Rpa190 (left panel) and Rpa135 (right panel) upon BMH-21 treatment is quantified relative to control condition (DMSO). Error bars indicate mean ± SD. \*\*\*p<0.005; \*\*p<0.01; \*p<0.05, calculated by two-sample t-test.

The online version of this article includes the following source data for figure 6:

**Source data 1.** Original pictures and blots used for *Figure 6A-C*.

**Source data 2.** Annotated pictures and blots used for *Figure 6A-C*.

**Source data 3.** Excel spreadsheet containing quantitative analysis of triplicates of *Figure 6C*.

ratio between distributions reaches twofold less accumulation in the presence of BMH-21, notably around the 100th nucleotide and between the 200th and the 230th nucleotide (*Figure 7A*).

CRAC data show that BMH-21 effects on Pol I elongation are specifically localized in regions where WT Pol I is highly accumulated, notably in the 5'ETS. This result suggests that BMH-21 specifically targets paused elongation complexes. Comparison of WT Pol I and SuperPol occupancy in the presence of BMH-21 shows that SuperPol occupancy is less affected by the drug (*Figure 7B*). We could show that SuperPol exhibits a decreased pausing relative to WT enzyme, which is consistent with the fact that SuperPol retains high levels of transcription upon BMH-21 treatment (see *Figure 6*).

In vitro transcription reaction experiments published by *Jacobs et al., 2022*, show that BMH-21 treatment increases pausing of the elongation complex and globally reduces the amount of Pol I elongation complexes capable of processive transcription. However, in vivo, both NET-seq and CRAC show decreased occupancy at the 5'end of rDNA, compatible with a putative massive premature termination in vivo.

We next measured the production of abortive rRNA in vivo, as presented earlier, by a northern blot using TSS probes with or without Rrp6 (*Figure 7C and D*). Cells were treated with 35 μM BMH-21 for 30 min. For the strain bearing WT Pol I, a very strong accumulation of short rRNA species between 70 and ≈ 300 nt long is detected (black arrows). The production of abortive transcripts is massively stimulated by BMH-21. Strains bearing SuperPol are less affected by the drug treatment, and the production of these abortive rRNA is mildly increased. This result suggests that BMH-21 stimulates premature termination of the WT Pol I.

Overall, these data suggest that BMH-21 targets Pol I-paused elongation complexes and stimulates their premature terminations, with production of abortive transcripts. As SuperPol results in a modified pausing pattern, less prone to premature termination, this mutant form is less affected by the drug and is resistant to its inhibitory effect on rRNA transcription elongation.

## Discussion

In our study, we characterized a mutant of Pol I, SuperPol, to better understand, by contrast, how rRNA production is regulated in vivo. We demonstrated that SuperPol overproduces rRNAs by reducing the rate of PTT in vivo. The enzymatic properties of SuperPol indeed result in reduced cleavage activity, improved processivity in vitro, but reduced fidelity. Finally, SuperPol is resistant to BMH-21 and retains high levels of transcription upon BMH-21 treatment. Therefore, we propose that BMH-21 acts on Pol I through targeting paused Pol I and stimulating premature termination. Overall, we propose that PTT by Pol I can be modulated. It can either be alleviated in the mutant, allowing increased rRNA production, or specifically enhanced by a drug, resulting in rapid inhibition of rRNA synthesis.

### Pol I is prone to PTT, and PTT is stimulated by BMH-21

PTT was first described in bacteria in the mid-1970s and was referred to as 'attenuation'. In mammals, Pol II PTT is now considered a major and widespread regulatory mechanism of the protein-coding genes at the transcriptional level. In this study, we have identified for the first time PTT during Pol I transcription in WT cells under normal growth conditions. We propose that stalled Pol I is inherently prone to premature termination. The occurrence of PTT during RNA Pol I transcription has previously

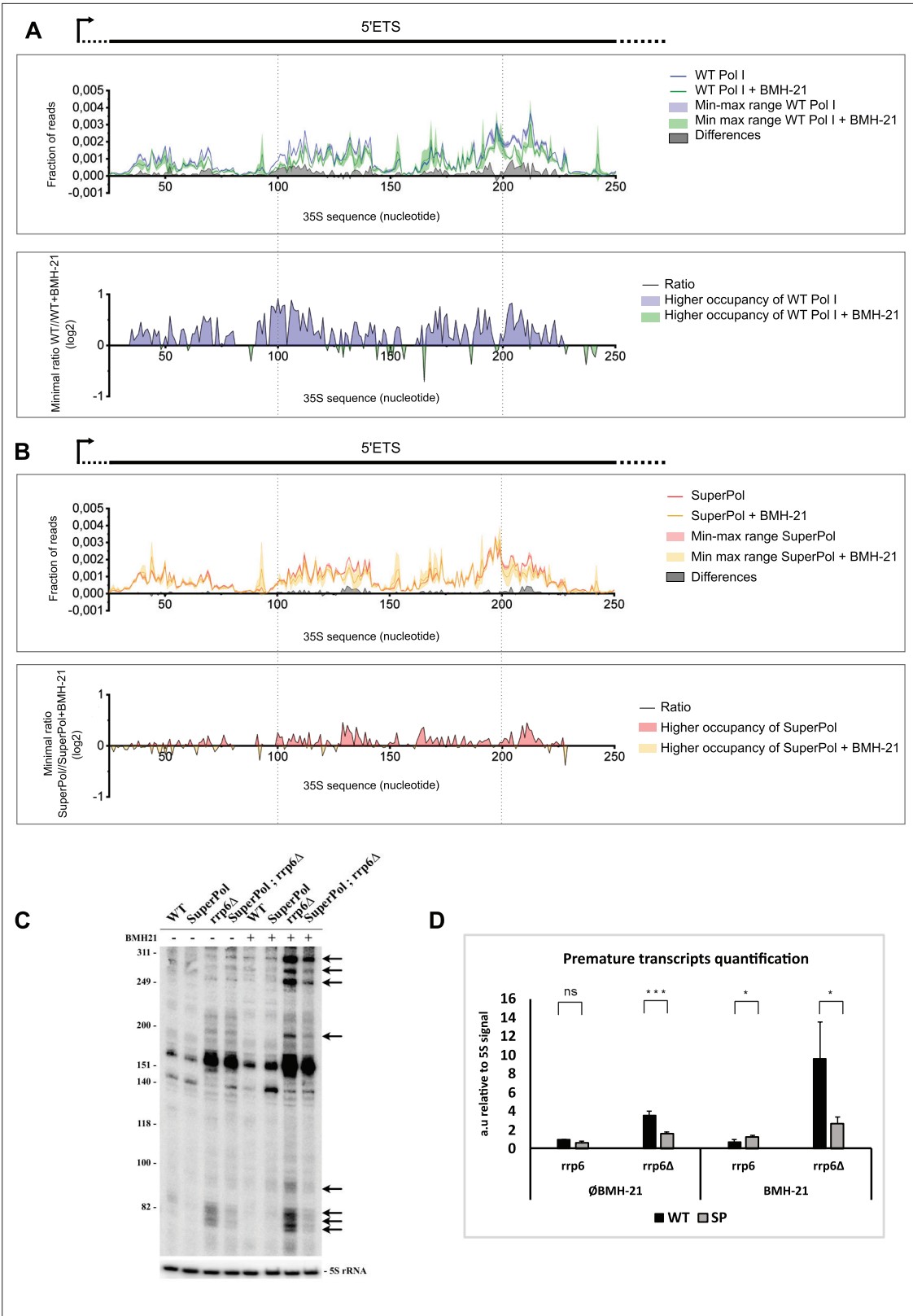

**Figure 7.** BMH-21 reduces polymerase I (Pol I) occupancy through targeting of paused Pol I and stimulation of premature termination. (**A**) Zoom-in 5′ external transcribed spacer (5′ETS) CRAC distribution profiles obtained for wild-type (WT) Pol I (blue) and WT Pol I incubated with 35 μM of BMH-21 for 30 min (green). In the upper panel, lines correspond to the mean frequency of reads obtained for two independent experiments, while the colored area corresponds to the min-max range of the two experiments. The gray area corresponds to the difference of frequency of reads at each position of the

*Figure 7 continued on next page*

*Figure 7 continued*

gene between WT Pol I and WT Pol I with BMH-21. This difference has been calculated using the min-max range values, giving the lowest difference (white area between blue and red min-max range). The lower panel represents the ratio between WT Pol I and WT Pol I with BMH-21, plotted on a log2 scale. Blue areas correspond to sequences where WT Pol I without BMH-21 is more accumulated, and green areas correspond to the sequences where WT Pol I incubated with BMH-21 is more accumulated. (**B**) Zoom-in 5'ETS CRAC distribution profiles obtained for SuperPol (Red) and SuperPol incubated with 35 µM of BMH-21 for 30 min (yellow). (**C**) Cells bearing WT Pol I or SuperPol, in the presence or absence of Rrp6, were grown and treated with 35 µM of BMH-21 for 30 min. Total RNAs were extracted and analyzed by northern blot using transcription start site (TSS) probe. Abortive transcripts are indicated by arrows. (**D**) Premature transcript products were quantified relative to 5S signal (focusing on ≈80 nt size). Error bars indicate mean ± SD. ***p<0.005; **p<0.01; *p<0.05, calculated by two-sample t-test.

The online version of this article includes the following source data for figure 7:

**Source data 1.** Original files used for *Figure 7C*.

**Source data 2.** Annotated Northern blots used for *Figure 7C*.

**Source data 3.** Excel spreadsheet containing the quantitative analysis of the CRAC experiment shown in *Figure 7*.

been reported only following DNA-binding drugs poisoning Pol I elongation (*Fetherston et al., 1984*; *Hadjiolova et al., 1995*; *Shcherbik et al., 2010*).

In our study, we propose that BMH-21, known to inhibit elongation, strongly stimulates premature termination. In vitro experiments conducted in Schneider's lab have shown that BMH-21 causes a delay in Pol I promoter escape efficiency, reduces Pol I elongation rate, and increases Pol I pausing (*Jacobs et al., 2022*). Using NET-seq, the authors observed a significant decrease in Pol I occupancy in the 5'ETS region, an observation fully compatible with our model. BMH-21 has been shown to lead to proteasome-dependent degradation of actively transcribing Pol I (*Wei et al., 2018*). In our study, we were able to substantiate these findings by demonstrating that BMH-21 treatment results in the degradation of Rpa190. Furthermore, we extended the initial discovery by revealing that both the largest subunit, Rpa190, and Rpa135 are destabilized by proteasome-dependent degradation, with no apparent effects on the other subunits. However, the precise mechanism behind the degradation of Rpa190 and Rpa135 remains to be clarified.

Recently, another Pol I allele resistant to BMH-21 was identified (*Ibars et al., 2023*). The Torres-Rosell's lab demonstrated that Nse1, an ubiquitin ligase subunit of the Smc5/6 complex, promotes ubiquitination of Rpa190 at positions K408 and K410. A double substitution of lysines at positions 408 and 410 with arginines, named Rpa190-KR, prevents ubiquitination of Rpa190 and confers resistance to BMH-21 (*Ibars et al., 2023*). Upon BMH-21 treatment, Rpa190-KR, similarly to SuperPol, shows resistance to proteasome-mediated degradation of both Rpa190 and Rpa135. Furthermore, SuperPol and Rpa190-KR have a synergistic effect on BMH-21 resistance (unpublished data), suggesting a potentially distinct mechanism. We now need to untangle the relationship between PTT stimulation, ubiquitination of Rpa190, and degradation to gain a better understanding of the specificity of BMH-21.

## Is transcriptional pausing coupled to PTT?

Pol II PTT is frequent, can occur near the TSS or within the gene body, and opposes the formation of full-length and functional transcripts, thereby negatively regulating gene expression. The release of the premature transcript from the elongating complex is thought to be preceded by a pause of the polymerase. The study of mammalian transcription by RNA Pol II suggested a concordant relationship between pause time and PTT: the longer the pause, the more frequent the release of Pol II from chromatin at the vicinity of the promoter (*Krebs et al., 2017*; *Steurer et al., 2018*). The timing of RNA Pol II pauses during transcription elongation plays a pivotal role in the regulation of gene expression. Extended pause durations can disrupt the transcription process, potentially resulting in incomplete mRNA formation or incorrect splicing events (for a review, see *Noe Gonzalez et al., 2021*). Pol II's residence time downstream of the TSS was investigated at a single-molecule level in *Drosophila*. Similar high Pol II turnover rates were observed downstream of promoters for both 'paused genes' and 'normally elongating genes' (*Krebs et al., 2017*). This observation demonstrates that Pol II, when stalled downstream of the promoter, is rapidly subjected to PTT with fast reinitiation cycles, rather than stable pausing. This interpretation is further supported by a study using fluorescence recovery after photobleaching in live human cells (*Steurer et al., 2018*). Computational modeling suggested that up to 90% of paused Pol II downstream of the promoter leads to PTT, with only 8% yielding a full-length mRNA. Therefore, the continuous release and reinitiation of Pol II in the vicinity of the

promoter are an important component of transcriptional regulation. These results strongly suggest a coupling between pause time and PTT: the longer the pause, the larger the fraction of Pol II released from chromatin.

Concerning Pol I, this aspect of regulation was not addressed so far. We propose that stalled RNA Pol I can be subjected to PTT rather than stable pausing. CRAC analysis (*Turowski et al., 2020* and this study), as well as EM studies of Miller chromatin spreads, demonstrated uneven distribution of polymerases along the rDNA gene, indicating different elongation rates due to differing pause durations. We propose that, at a given position, the duration of the pause could regulate the fate of the Pol I elongation complex. A Pol I mutant with an increased pausing frequency and a reduced transcriptional elongation rate (*Viktorovskaya et al., 2013*) has been associated with rRNA processing defects, resulting in an impaired ribosome biogenesis (*Huffines et al., 2022*). This phenotype may be the consequence of an increased rate of PTT.

Conversely, the SuperPol mutant, which exhibits a lower pausing rate, shows reduced levels of PTT. As a result, this mutant is capable of producing 1.5 times more rRNAs than the WT. It is worth noting that no processing defect (*Darrière et al., 2019*) nor co-transcriptional modification defect (ribomethSeq, data not shown) was observed in the SuperPol mutant when the integrity of the exosome is maintained. This suggests that putative aberrant pre-rRNAs produced consecutively to shorter pauses do not accumulate as long as the quality control mechanisms are efficient. In summary, overall fine-tuning pausing time may be necessary to strike a compromise between slow but inefficiently productive elongation and fast but error-prone elongation, respectively.

## What could be the biological relevance of Pol I pausing associated with PTT?

The average Pol I elongation rate is significantly reduced throughout the 5' external transcribed spacer (ETS) region, where both massive early pre-rRNA assembly events and frequent PTT occur. The amount of protein bound to the 5' ETS corresponds to 40% of the total mass of the 90S preribosomal particle (*Sun et al., 2017*). Furthermore, studies in yeast have demonstrated that the 5' region of rRNA can pull down and cross-link the UTP-A complex (*Hunziker et al., 2016*; *Zhang et al., 2016*). The recruitment of the UTP-A complex is of paramount importance as it is a prerequisite for the assembly of all other small-subunit processome subcomplexes. We propose that Pol I pausing in the first 300 nt downstream of the TSS offers a window of opportunity for quality control: the UTP-A assembly onto the nascent transcript. Utp9, Utp8, and Utp17 are the UTP-A components recruited closest to the 5' end, within the first 60 nt of the 5' ETS. We speculate that the resumption of Pol I elongation from this point depends on the assembly of these three proteins. In the absence of UTP-A binding, extended pausing would stimulate PTT and the production of an ≈80-nt-long abortive transcript. Similarly, Utp4, Utp5, Utp10, and Utp15 associate later to 5'ETS, up to position +350 and we also detect abortive transcripts in this range, suggesting similar quality control points in this area. This speculative model is supported by the fact that UTP-A components, notably Utp5, Utp15, and Utp17, are required for optimal rDNA transcription, as observed in run-on analyses (*Gallagher et al., 2004*).

## Conclusion

In conclusion, the control of pause durations in RNA Pol II and RNA Pol I transcription elongation is of utmost importance for precise gene expression regulation. Excessive prolongation of pause durations can lead to transcriptional deregulation, while the strict regulation of pause durations ensures accurate transcription. Understanding the dynamics of pause durations in transcriptional processes is crucial for unraveling the intricacies of gene expression.

## Limitations of the study

In experiments presented here, we likely underestimated the occurrence of PTT for Pol I. Our study established PTT by focusing on the production of abortive transcript, which was detected only in the absence of Rrp6. The novel implication of a torpedo-like mechanism of premature release in 5'ETS and 3'ETS, as proposed by *Petfalski et al., 2025*, strongly suggests that PTT is also occurring following Rat1-dependent degradation of nascent rRNA. Taken together, PTT associated with transcript release, or coupled with the torpedo mechanism, is likely to represent a prominent regulatory step in the transcription cycle of rDNA.

## Materials and methods

### Strains, media, plasmids, and cloning

Propagation of yeast was performed using standard rich YP medium (1% yeast extract, 2% peptone) supplemented with either 2% glucose (YPD) or 2% galactose (YPG) or using minimal YNB medium (0.67% yeast nitrogen base, 0.5% $(NH_4)_2SO_4$ and 2% glucose or galactose) supplemented with the required amino acids. Yeast strains used in this study were derivatives of *S. cerevisiae* BY4741, originating from the S288C background. Strains 3207 and 3208 were constructed as follows: a PCR cassette containing the HTP-tag sequence followed by the kanamycin resistance (kanR) selectable marker was amplified by PCR from the plasmid pBS-1539-HTP (*Granneman et al., 2009*) using primers 1898 and 1899. The PCR fragment was inserted by homologous recombination downstream of the RPA135 and RPC128 chromosomal open reading frame in the BY4742 strain. Transformants were selected for uracil prototrophy, screened by immunoblotting, and sequenced. Deletion of *RRP6* was performed as previously described (*Dez et al., 2006*).

### Pol I TMA

Yeast cells were grown in phosphate-depleted YPD medium (*Briand et al., 2001*) to an OD600 of 0.8 at 30°C. The RNAs were labeled in vivo by incubation of 1 ml culture aliquots with 150 μCi [$^{32}$P]orthophosphate (P-RB-1, (54mCi/ml) Hartmann Analytic, Braunschweig, Germany) for the indicated time. Cells were collected by centrifugation, and the pellets were frozen in liquid nitrogen. RNAs were then extracted (*Beltrame and Tollervey, 1992*) and precipitated with ethanol. Slot blots were loaded with single-stranded 80-mers DNA oligonucleotides: 1855 (NTS2), 1856 (5′ETS-1), 1857 (5′ETS-2), 1858 (5′ETS-3), 1859 (18S.2), 1860 (25S.1), 1861 (3′ETS), 1863 (5S US), and 1864 (5S DS), and hybridization was performed as previously described for TRO (*Prescott et al., 2004*; *Wery et al., 2009*).

### Western analyses

Proteins from total extracts obtained after TCA precipitation or from immunoprecipitated pellets were separated on 4–12% polyacrylamide/SDS gels (Bio-Rad) and transferred to Hybond-ECL membranes (GE Healthcare). HTP-tagged proteins were detected using rabbit PAP (Sigma) diluted 10,000-fold. Pol I subunits were detected using polyclonal anti-Pol I total antibody as previously described (*Riva et al., 1982*). Swi6 was detected using polyclonal anti-Swi6 antibody (Cusabio, PA362844XA01SVG).

### RNA extractions and northern hybridizations

RNA extractions and northern hybridizations were performed as previously described (*Beltrame and Tollervey, 1992*). Low-molecular-weight RNA products were resolved on 8% polyacrylamide/8.3 M urea gels.

### CRAC experiment

Samples were processed as previously described (*Turowski et al., 2020*).

#### Culture

Yeast cells expressing Rpa135 and/or Rpc128 fused at the C-terminus to the tripartite HTP tag (His6 tag-TEV protease cleavage site-Protein A tag) and the WT BY4741 strain (negative control) were grown at 30°C in 2.5 l of synthetic dextrose medium with 2% glucose, lacking tryptophan, to an OD600 of 0.9. When indicated, 35 μM of BMH-21 was added to the culture for 15 min before UV irradiation.

#### UV cross-linking, cell lysis, supernatant clarification

Cells were irradiated with a megatron for 100 s with UV light at 254 nm and harvested. Cells were resuspended in TNMC100 buffer (50 mM Tris-HCl pH 7.5, 150 mM NaCl, 0.1% NP-40, 5 mM MgCl$_2$, 10 mM CaCl$_2$, 5 mM β-mercaptoethanol, 50 U of DNase RQ1, and a protease-inhibitor cocktail-1 tablet/50 ml) and lysed by mechanical disruption using zirconia beads with six 1 min pulses, with cooling on ice in between. The supernatant was spun for 20 min at 13,000 rpm.

## IgG sepharose incubation+washes+TEV elution

Cell lysates were mixed with 400 μl IgG Sepharose 6 Fast Flow slurry (GE Healthcare) pre-equilibrated with TNM100 buffer (50 mM Tris-HCl pH 7.8, 100 mM NaCl, 0.1% NP-40, 5 mM MgCl$_2$, 5 mM β-mercaptoethanol) and incubated for 2 hr at 4°C on a stirring wheel. Beads were washed two times with TNM600 buffer (50 mM Tris-HCl pH 7.8, 600 mM NaCl, 1.5 mM MgCl$_2$, 0.1% NP-40, 5 mM β-mercaptoethanol) and two times with TNM100 buffer, then resuspended in 600 μl TNM100 buffer and transferred into Micro Bio-Spin 6 columns (Bio-Rad). The elution from IgG Sepharose beads was achieved by incubation with 30 μl homemade GST-tagged TEV protease for 2 hr on a shaking table at 16°C.

## Digestion of TEV eluates+Ni-NTA incubation+washes

TEV eluates (about 650–700 μl) were partially digested for 5 min at 37°C with 25 μl of RNase-IT (Agilent) diluted to 1:50 in TNM100 buffer, and the reactions were stopped using 0.4 g guanidine hydrochloride. The resulting samples were supplemented with 300 mM NaCl and 10 mM imidazole and incubated overnight at 4°C on a stirring wheel with 50 μl Ni-NTA agarose resin slurry (QIAGEN) pre-equilibrated with wash buffer I (50 mM Tris-HCl pH 7.8, 300 mM NaCl, 10 mM imidazole, 6 M guanidine hydrochloride, 0.1% NP-40, 5 mM β-mercaptoethanol). Ni-NTA beads were then washed two times with wash buffer I, three times with 1× PNK buffer (50 mM Tris-HCl pH 7.8, 10 mM MgCl$_2$, 0.5% NP-40, 5 mM β-mercaptoethanol) and transferred into Mobicol Spin Columns.

## 3'miRCat-33 ligation

The miRCat-33 3' linker was ligated to the 3' end of the RNAs on the Ni-NTA beads with 800 units of T4 RNA ligase 2 truncated K227Q (New England Biolabs) in 1× PNK buffer/16.67% PEG 8000 in the presence of 80 units RNasin in a total volume of 80 μl. The ligation reaction was incubated for 5 hr at 25°C. Beads were washed once with wash buffer I to inactivate the RNA ligase and three times with 1× PNK buffer.

## 5' phosphorylation

The 5' ends of the RNAs were then radiolabeled by phosphorylation in reactions containing 1× PNK buffer, 40 μCi of 32P-γ ATP, and 20 units of T4 PNK (Sigma) in a total volume of 80 μl. The reactions were incubated at 37°C for 40 min. To ensure all RNAs get phosphorylated at the 5' end for downstream ligation of the 5' linker, 1 μl of 100 mM ATP was added to the reaction mix, which was incubated for another 20 min at 37°C. Beads were washed once with wash buffer I to inactivate the kinase and four times with 1× PNK buffer.

## 5' linker ligation

5' adaptor was ligated to the 5' end of the RNAs retained on the Ni-NTA beads. Ligation reactions contained 1× PNK buffer, 1.25 μM of 5' adaptor, 1 mM ATP, 40 units of T4 RNA Ligase 1 (New England Biolabs) in a total volume of 80 μl. The reactions were incubated overnight at 16°C. Beads were washed three times with wash buffer II (50 mM Tris-HCl [pH 7.8], 50 mM NaCl, 10 mM imidazole, 0.1% NP-40, 5 mM β-mercaptoethanol).

## TCA precipitation of the eluate

The material retained on the beads was then eluted using two times 200 μl wash buffer II (50 mM Tris-HCl [pH 7.8], 50 mM NaCl, 150 mM imidazole, 0.1% NP-40, 5 mM β-mercaptoethanol). Eluates were precipitated with TCA (20% final concentration) in the presence of 30 μg of glycogen (Roche) to favor precipitation. The precipitated material was resuspended in NuPAGE™ LDS sample buffer (Invitrogen) with reducing agent (Invitrogen), heated 10 min at 65°C, loaded on NuPAGE™ 4–12% Bis-Tris gels (Invitrogen) and run in 1× MOPS SDS running buffer (Invitrogen). The material was then transferred onto Amersham Protran Nitrocellulose Blotting Membrane (GE Healthcare) using a transfer buffer containing 1× NuPAGE (Invitrogen) and 20% MeOH, for 2 hr at 25 V and 4°C. The area of the membrane containing a radioactive signal at the expected size of Rpa135 protein was excised. A membrane area at the same size was excised in the BY4741 sample lane. Membranes were soaked in 400 μl wash buffer II supplemented with 1% SDS, 5 mM EDTA, and proteins were degraded using 100 μg proteinase K (Sigma) and incubation for 2 hr at 55°C. RNA was extracted with

phenol:chloroform:isoamyl alcohol (25:24:1) and then precipitated by addition of 1:10 volume of 3 M sodium acetate (pH 5.2), 2.5 volumes of 100% ethanol, and 20 µg of glycogen. Dried RNA pellets were dissolved in ultrapure MilliQ $H_2O$.

## Reverse transcription+PCR amplification

Synthesis of cDNAs was performed using SuperScript III reverse transcriptase (Thermo Fisher Scientific) and 'RT primer' oligonucleotide (see Materials and methods). The resulting cDNAs were PCR-amplified using LA Taq DNA polymerase (TaKaRa), a common PCR primer used for all library amplifications, (RP1) and a relevant specific index primer for multiplexing: RPI1, RPI2, RPI3, RPI4, RPI5, and RPI6.

Note that a first set of experiments was performed using L5Aa, L5Ab, and L5Ac 5' linkers. 'RT primer' was used for RT reactions, and PCRs were performed using P5F and P3R oligonucleotides.

## PCR product purification

The resulting PCR products were purified by phenol:chloroform:isoamyl alcohol extraction and ethanol precipitation. After agarose gel electrophoresis (agarose 'small fragments', Eurogentec) run in 1× TBE buffer and stained with SYBR Safe DNA gel stain (Invitrogen), DNA fragments ranging in size between 150 and 250 base pairs were gel-purified using MinElute PCR Purification Kit (QIAGEN). Concentration of the final DNA samples was measured using Qubit dsDNA HS Assay Kit (Invitrogen) and a Qubit fluorometer (Thermo Fisher Scientific), and the samples were sent to the Epitranscriptomics & Sequencing (EpiRNA-Seq) facility for Illumina sequencing.

## Deep-sequencing and computational analyses

Sequencing was performed using an Illumina MiSeq system. Adapters and low-quality reads were eliminated using Flexbar Reads UMI barcode was extracted using umi_tools version 1.1.2. [OGT1] Reads were cleared of their adapter with cutadapt version 1.18, and cleaned reads were aligned to the yeast genome both on the entire genome version EF2.59.1 (R63-1-1) and on the rDNA sequence using Novoalign version 3.09.00. SAM files were converted (view -b), completed (fixmate), ordered (sort) using samtools version 1.14 PCR duplicates were then removed using a custom script in R, based on both start, end, and umi barcode. Quality control was applied at each step using fastQC version 0.11.9 and samtools flagstat. Per base coverage based on the 3' end of the reads on the different alignments was generated using R and in-house scripts available on our GitHub. Downstream analyses, including the pileups, were performed using the pyCRAC tool suite. Hits repartition per million of sequences was produced using pyReadCounter.py—m 1,000,000 option. Different pileups of hits for each gene were obtained using pyPileup.py—L 50—limit = 100,000 options. Raw and processed data are available in the Gene Expression Omnibus database under the accession number GSE247803. The entire pipeline, developed in Snakemake using conda environment, including our in-house is available on https://github.com/CBIbigA/crac-seq/ (*Rocher, 2024*).

## **In vitro transcriptional assays**

### Pol I purification

RNA Pol I was purified as described in *Schwank et al., 2022*, with the following modifications. Y4006, Y4007, and Y4089 strains were grown in YPD medium (2% [wt/vol] peptone, 2% [wt/vol] glucose, 1% [wt/vol] yeast extract) and harvested at $OD_{600}$~2.0. Cells were resuspended in lysis buffer (1.5 ml lysis buffer per 1 g cells) in cold Zirconia tubes (6 ml per 12 g glass beads Ø 0.75–1 mm) and mechanically lysed with Precellys Evolution (Bertin Technologies) using 6× pulse at 6000 rpm for 30 s followed by 30 s pause in between. The extract was incubated with IgG sepharose beads that have been pre-equilibrated in lysis buffer on a turning wheel for 2 hr at 4°C (1 µl bead suspension per 2 mg protein). Beads were washed four times with 1 ml of wash buffer (20 mM HEPES/KOH [pH 7.8], 20% [vol/vol] glycerol, 5 mM $MgCl_2$, 1500 mM KOAc, and 0.15% [vol/vol] NP-40) for 10 min at 4°C and equilibrated three times with elution buffer (20 mM HEPES/KOH [pH 7.8], 10% [vol/vol] glycerol, 5 mM $MgCl_2$, 200 mM KOAc, and 5 µM $ZnCl_2$) for 3 min at 4°C. Beads were resuspended in elution buffer (1 µl elution buffer per 2 µl beads suspension), incubated with TEV protease overnight at 16°C and eluate

is recovered and analyzed via SDS-PAGE and Coomassie staining and is used for the in vitro transcriptional assays.

## Template generation

The minimal transcription templates used in this study are produced as described in *Schwank et al., 2022*. The oligonucleotides for template generation are listed in Materials and methods. The RNA-DNA scaffold was diluted to 0.1 μM for subsequent RNA cleavage or elongation assays.

## In vitro transcriptional assays

RNA cleavages assay was conducted as described in *Schwank et al., 2022*, with 0.2 pmol of Pol I incubated with 0.066 pmol of the respective preannealed minimal transcription cleavage scaffold in 1× transcription buffer (20 mM HEPES/KOH [pH 7.8], 10 mM $MgCl_2$, 5 mM EGTA/KOH [pH 8.0], 5 μM $ZnCl_2$, and 100 mM KOAc) at 4°C for 10–20 min. For the elongation assay, samples were implemented with NTPs (200 μM final concentration of each). Samples are incubated at 28°C for 1–30 min. The reaction was stopped using an equal amount of 2× TBE loading dye (8 M urea, 0.08% [wt/vol] bromophenol blue, 4 mM EDTA/NaOH [pH 8.0], 178 mM Tris-HCl, and 178 mM boric acid). The 5'Cy5-labeled RNA was size-separated on a 20% denaturing polyacrylamide gel (20% [wt/vol] acrylamide/bisacrylamide [19:1], 6 M urea, 0.1% [vol/vol] *N*,N,*N*',*N*'-tetramethylethylenediamine, 0.1% [wt/vol] ammonium persulfate, 2 mM EDTA/NaOH [pH 8.0], 89 mM Tris-HCl, and 89 mM boric acid) using 1× TBE running buffer (89 mM Tris-HCl, 89 mM boric acid, and 2 mM EDTA/NaOH [pH 8.0]) and visualized with a Typhoon imaging system (GE Healthcare).

## Strains used in this study

| Strain | Name | Genotype | Plasmid | Origin or reference |
|---|---|---|---|---|
| 0480 | BY4741 | *MAT a, his3Δ1, leu2Δ0, lys2Δ0 ura3Δ0* | | This study |
| 2742 | OGT32-1a | *MAT a, his3Δ1, leu2Δ0, lys2Δ0 ura3Δ0, RPA135-F301S::URA3* | | This study |
| 2743 | yCD2-1a | *MAT a, his3Δ1, leu2Δ0, lys2Δ0 ura3Δ0, rrp6::NAT* | | This study |
| 2744 | OGT33-1a | *MAT a, his3Δ1, leu2Δ0, lys2Δ0 ura3Δ0, RPA135-F301S::URA3, rrp6::NAT* | | This study |
| 3207 | yCA3-1a | *MAT a, his3Δ1, leu2Δ0, lys2Δ0 ura3Δ0, RPA135-HTP::URA3* | | This study |
| 3208 | yCA2-1a | *MAT a, his3Δ1, leu2Δ0, lys2Δ0 ura3Δ0, RPA135-F301S-HTP::URA3* | | This study |
| | Y4006 | MATa his3Δ1 leu2Δ0 lysΔ0 ura3Δ0 RPA135-F301S::URA3 KANMX-pGAL::RPA12 | pCM182-LEU2-RPA12 | *Schwank et al., 2022* |
| | Y4007 | MATa his3Δ1 leu2Δ0 lysΔ0 ura3Δ0 RPA135-F301S::URA3 KANMX-pGAL::RPA12 | pCM182-LEU2-rpa12delC (aa 1–69) | *Schwank et al., 2022* |

## Oligonucleotides used in this study

| Name | 5'–3' Sequence |
|---|---|
| 774 | gtcttcaactgctttcgcat |
| 1855 | aaatggcctatcggaatacattttctacatcctaactactata aaacaacctttagacttacgtttgctactctcatggt |
| 1856 | TACAAAAACATAACGAACGACAAGCCTAC TCGAATTCGTTTCCAAACTCTTTTCGAACTTGT CTTCAACTGCTTTCGCAT |
| 1857 | tgcgaccggctattcaacaaggcattcccccaagtttgaattctttga aatagattgctattagctagtaatccaccaaa |
| 1858 | TATCTTAAAAGAAGAAGCAACAAGCAGTAAAA AAGAAAGAAACCGAAATCTCTTTTTTTTTT TCCCACCTATTCCCTCTT |
| 1859 | ggaattcctcgttgaagagcaataattacaatgctctatccccagcac gacggagtttcacaagattaccaagacctctc |

*Continued on next page*

*Continued*

| Name | 5'–3' Sequence |
|---|---|
| 1860 | gtgctggcctcttccagccataagaccccatctccggataaaccaatt ccggggtgataagctgttaagaagaaaagata |
| 1861 | gtaaatggtacactcttacacactatcatcctcatcgtatattataata gatatatacaatacatgtttttacccggatc |
| 1863 | cagcttaactacagttgatcggacgggaaacggtgctttctggtagat atggccgcaaccgatagtttaacggaaacgca |
| 1864 | aaaaaaaaaaaagaaataaagattgcagcacctgagtttcgcgtatg gtcacccactacactactcggtcaggctcttac |
| 1898 | CCTCTATATGATACAATTGACCAAGCCTTCATTT ACCATTCTATATCAATTTGGAAAGAAGGG TATTTCTTACGACTCACTATAGGG |
| 1899 | TGAAGTACTTGGACTCTGAGCTATCCGGCAATG GGTATAAGATTGCGTTATAATGTAGAGCC CAAAGAGCACCATCACCATCACCATGATTATGATATTCC |
| miRCat-33 3 ' | rAppTGGAATTCTCGGGTGCCAAG/ddC/ |
| 5' adaptor | InvddT-GTTCAGAGTTCTACAGTCCGACGATCNNNNNAGC-OH |
| RT primer | GCCTTGGCACCCGAGAATTCCA |
| RP1 | AATGATACGGCGACCACCGAGATCTACACGTTC AGAGTTCTACAGTCCGA |
| RPI4 | CAAGCAGAAGACGGCATACGAGATTGGTCAGTGACTGG AGTTCCTTGGCACCCGAGAATTCCA |
| RPI5 | CAAGCAGAAGACGGCATACGAGATCACTGTGTGACTGG AGTCCTTGGCACCCGAGAATTCCA |
| RPI6 | CAAGCAGAAGACGGCATACGAGATATTGGCGTGACTGG AGTTCCTTGGCACCCGAGAATTCCA |
| RPI7 | CAAGCAGAAGACGGCATACGAGATGATCTGGTGACTGG AGTTCCTTGGCACCCGAGAATTCCA |
| RPI12 | CAAGCAGAAGACGGCATACGAGATTACAAGGTGACTGG AGTTCCTTGGCACCCGAGAATTCCA |
| L5Aa | invddT-ACACrGrArCrGrCrUrCrUrUrCrCrGrArUrCrUrNrNrUrArArGrC-OH |
| L5Ab | invddT-ACACrGrArCrGrCrUrCrUrUrCrCrGrArUrCrUrNrNrArUrUrArGrC-OH |
| L5Ac | invddT-ACACrGrArCrGrCrUrCrUrUrCrCrGrArUrCrUrNrNrNrGrCrGrCrArGrC-OH |
| P5F | AATGATACGGCGACCACCGAGATCTACACTCTTTCCCTAC ACGACGCTCTTCCGATCT |
| P3R | CAAGCAGAAGACGGCATACGAGATCCTTGGCACCCGAGAATTCC |
| 4634 | AGCTCAAGTACTTTTTCCTGGACATTCGCACCACGCGT |
| 4635 | TAAGTACTTGAGCT |
| 4633 | Cy5-UGCAUUUCGUCCAGGACGU |
| 4821 | Cy5-UGCAUUUCGUCCAGGAC |

## Materials availability statement

All materials generated in this study are available from the corresponding author upon reasonable request and may be subject to material transfer agreements.

## Acknowledgements

We are very grateful to our team members for critical reading of the manuscript and Vincent Rocher for bioinformatic analysis. We thank the 'Ligue Nationale Contre le Cancer' for supporting CA. We acknowledge the members of the Gadal lab for help, advice, and discussion. We are very grateful to Virginie Marchand and Iouri Motorine for the EpiRNA-Seq facility in Nancy for the help and sequencing

CRAC experiments. This work also benefited from the assistance of the Big-A facility of the CBI for the process and the analysis of CRAC data. This work was supported by the 'Fondation ARC pour la Recherche sur le Cancer' (ARCPJA 20191209547 and ARCPJA22020060002067) and the 'Ligue Nationale Contre le Cancer' (3FI14194UPAL). CA was funded by a Ph.D. fellowship from the 'Ligue Nationale Contre le Cancer' (TAGF23235).

## Additional information

### Funding

| Funder | Grant reference number | Author |
|---|---|---|
| Fondation ARC pour la Recherche sur le Cancer | ARCPJA 20191209547 | Olivier Gadal |
| Fondation ARC pour la Recherche sur le Cancer | ARCPJA22020060002067 | Christophe Dez |
| Ligue Contre le Cancer | 3FI14194UPAL | Christophe Dez |
| Ligue Contre le Cancer | TAGF23235 | Chaïma Azouzi |
| Deutsche Forschungsgemeinschaft | TS35/15-1 695545 | Herbert Tschochner |

The funders had no role in study design, data collection and interpretation, or the decision to submit the work for publication.

### Author contributions

Chaïma Azouzi, Data curation, Formal analysis, Methodology, Writing – original draft; Katrin Schwank, Marta Kwapisz, Investigation, Methodology; Sophie Queille, Methodology; Marion Aguirrebengoa, Simon Lebaron, Software, Methodology; Anthony Henras, Formal analysis, Methodology; Herbert Tschochner, Conceptualization, Supervision, Validation, Methodology; Annick Lesne, Formal analysis; Frederic Beckouët, Supervision, Investigation; Olivier Gadal, Conceptualization, Resources, Data curation, Software, Formal analysis, Supervision, Funding acquisition, Validation, Investigation, Visualization, Methodology, Writing – review and editing; Christophe Dez, Conceptualization, Resources, Data curation, Formal analysis, Supervision, Funding acquisition, Validation, Investigation, Visualization, Methodology, Writing – original draft, Project administration, Writing – review and editing

### Author ORCIDs

Chaïma Azouzi https://orcid.org/0000-0002-8816-7862
Sophie Queille https://orcid.org/0009-0007-1172-9883
Marta Kwapisz https://orcid.org/0009-0008-9900-4103
Marion Aguirrebengoa https://orcid.org/0000-0001-5205-9774
Anthony Henras https://orcid.org/0000-0001-7785-9938
Simon Lebaron https://orcid.org/0000-0001-7348-2088
Herbert Tschochner https://orcid.org/0000-0002-3560-861X
Annick Lesne https://orcid.org/0000-0002-6647-612X
Frederic Beckouët https://orcid.org/0000-0002-9009-1852
Olivier Gadal https://orcid.org/0000-0001-9421-0831
Christophe Dez https://orcid.org/0000-0002-8526-3541

Reviewer #1 (Public review): https://doi.org/10.7554/eLife.106503.3.sa1
Reviewer #2 (Public review): https://doi.org/10.7554/eLife.106503.3.sa2
Reviewer #3 (Public review): https://doi.org/10.7554/eLife.106503.3.sa3
Author response https://doi.org/10.7554/eLife.106503.3.sa4

## Additional files

### Supplementary files
MDAR checklist

### Data availability
DNA sequence data were deposited in GEO (accession number GSE247803). All data generated or analyzed during this study are included in the manuscript and supporting files; source data files have been provided for all figures.

The following dataset was generated:

| Author(s) | Year | Dataset title | Dataset URL | Database and Identifier |
|-----------|------|---------------|-------------|-------------------------|
| Azouzi C, Aguirrebengoa M, Dez C, Gadal O | 2025 | Ribosomal RNA synthesis by RNA polymerase I is regulated by premature termination of transcription | https://www.ncbi.nlm.nih.gov/geo/query/acc.cgi?acc=GSE247803 | NCBI Gene Expression Omnibus, GSE247803 |

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
