## [Editor Report · eLife Assessment]

This manuscript characterizes a mutated clone of RNA polymerase I in yeast, referred to as SuperPol, to understand the mechanisms of RNA polymerase I elongation and termination. The authors present **convincing** evidence that demonstrates the existence of premature termination in Pol I transcription. Overall, the characterization of this RNA pol I offers **important** insights into the regulation of ribosomal RNA transcription and its potential application in cancer pharmacology.

[Editors' note: this paper was reviewed by Review Commons.]

---

## [Referee Report · Reviewer #1 (Public review)]

Summary:

The study characterises an RNA polymerase (Pol) I mutant (RPA135-F301S) named SuperPol. This mutant was previously shown to increase yeast ribosomal RNA (rRNA) production by Transcription Run-On (TRO). In this work, the authors confirm this mutation increases rRNA transcription using a slight variation of the TRO method, Transcriptional Monitoring Assay (TMA), which also allows the analysis of partially degraded RNA molecules. The authors show a reduction of abortive rRNA transcription in cells expressing the SuperPol mutant and a modest occupancy decrease at the 5' region of the rRNA genes compared to WT Pol I. These results suggest that the SuperPol mutant displays a lower frequency of premature termination. Using in vitro assays, the authors found that the mutation induces an enhanced elongation speed and a lower cleavage activity on mismatched nucleotides at the 3' end of the RNA. Finally, SuperPol mutant was found to be less sensitive to BMH-21, a DNA intercalating agent that blocks Pol I transcription and triggers the degradation of the Pol I subunit, Rpa190. Compared to WT Pol I, short BMH-21 treatment has little effect on SuperPol transcription activity, and consequently, SuperPol mutation decreases cell sensitivity to BMH-21.

Significance:

The work further characterises a single amino acid mutation of one of the largest yeast Pol I subunits (RPA135-F301S). While this mutation was previously shown to increase rRNA synthesis, the current work expands the SuperPol mutant characterisation, providing details of how RPA135-F301S modifies the enzymatic properties of yeast Pol I. In addition, their findings suggest that yeast Pol I transcription can be subjected to premature termination in vivo. The molecular basis and potential regulatory functions of this phenomenon could be explored in additional studies.

Our understanding of rRNA transcription is limited, and the findings of this work may be interesting to the transcription community. Moreover, targeting Pol I activity is an open strategy for cancer treatment. Thus, the resistance of SuperPol mutant to BMH-21 might also be of interest to a broader community, although these findings are yet to be confirmed in human Pol I and with more specific Pol I inhibitors in future.

Comments on revision:

The authors' response addressed all the points I raised adequately.

---

## [Referee Report · Reviewer #2 (Public review)]

Summary:

This article presents a study on a mutant form of RNA polymerase I (RNAPI) in yeast, referred to as SuperPol, which demonstrates increased rRNA production compared to the wild-type enzyme. While rRNA production levels are elevated in the mutant, RNAPI occupancy as detected by CRAC is reduced at the 5' end of rDNA transcription units. The authors interpret these findings by proposing that the wild-type RNAPI pauses in the external transcribed spacer (ETS), leading to premature transcription termination (PTT) and degradation of truncated rRNAs by the RNA exosome (Rrp6). They further show that SuperPol's enhanced activity is linked to a lower frequency of PTT events, likely due to altered elongation dynamics and reduced RNA cleavage activity, as supported by both in vivo and in vitro data.

The study also examines the impact of BMH-21, a drug known to inhibit Pol I elongation, and shows that SuperPol is less sensitive to this drug, as demonstrated through genetic, biochemical, and in vivo approaches. The authors show that BMH-21 treatment induces premature termination in wild-type Pol I, but only to a lesser extent in SuperPol. They suggest that BMH-21 promotes termination by targeting paused Pol I complexes and propose that PTT is an important regulatory mechanism for rRNA production in yeast.

The data presented are of high quality and support the notion that (1) premature transcription termination occurs at the 5' end of rDNA transcription units; (2) SuperPol has an increased elongation rate with reduced premature termination; and (3) BMH-21 promotes both pausing and termination. The authors employ several complementary methods, including in vitro transcription assays. These results are significant and of interest for a broad audience.

Adding experiments in different growth conditions to support the claim of regulation by PTT (as the authors propose) will also be an important addition. The revisions further support the claim, with in particular the notion that increased elongation rate of superpol occurs at the expense of fidelity.

Significance:

These results are significant and of interest for a basic research audience.

---

## [Referee Report · Reviewer #3 (Public review)]

In the manuscript "Ribosomal RNA synthesis by RNA polymerase I is regulated by premature termination of transcription", Azouzi and co-authors investigate the regulatory mechanisms of ribosomal RNA (rRNA) transcription by RNA Polymerase I (RNAPI) in the budding yeast *S. cerevisiae*. They follow up on exploring the molecular basis of a mutant allele of the second-largest subunit of RNAPI, RPA135-F301S, also dubbed SuperPol, that they had previously reported (Darrière et al, 2019), and which was shown to rescue Rpa49-linked growth defects, possibly by increasing rRNA production.

Through a combination of genomic and in vitro approaches, the authors test the hypothesis that RNAPI activity could be subjected to a premature transcription termination (PTT) mechanism, akin to what is observed for RNA Polymerase II (RNAPII). The authors demonstrate that SuperPol increased processivity "desensitizes" RNAPI to abortive transcription cycles at the expense of decreased fidelity. In agreement, SuperPol is shown to be resistant to BMH-21, a drug previously shown to impair RNAPI elongation.

Overall, this work expands the mechanistic understanding of the early dynamics of RNAPI transcription. The presented results are of interest for researchers studying transcription regulation, particularly those interested in RNAPI's transcription mechanisms and fidelity.

Strengths:

Overall, the experiments are performed with rigor and include the appropriate controls and statistical analyses. Conclusions are drawn from appropriate experiments. Both the figures and the text present the data clearly. The Materials and Methods section is detailed enough.

Weaknesses:

The biological significance of this phenomenon remains unaddressed and thus unclear. The lack of experiments to test a specific regulatory function (such as UTP-A loading checkpoint or other mechanisms) limit these termination events to possibly abortive actions of unclear significance.

Comments on revised version:

I appreciated the additional experiments and the other changes made by the authors in the revised version.

---

## [Author Response]

The following is the authors’ response to the original reviews

**General Statements:**

In our manuscript, we demonstrate for the first time that RNA Polymerase I (Pol I) can prematurely release nascent transcripts at the 5' end of ribosomal DNA transcription units in vivo. This achievement was made possible by comparing wild-type Pol I with a mutant form of Pol I, hereafter called SuperPol previously isolated in our lab (Darrière at al., 2019). By combining in vivo analysis of rRNA synthesis (using pulse-labelling of nascent transcript and cross-linking of nascent transcript - CRAC) with in vitro analysis, we could show that Superpol reduced premature transcript release due to altered elongation dynamics and reduced RNA cleavage activity. Such premature release could reflect regulatory mechanisms controlling rRNA synthesis. Importantly, This increased processivity of SuperPol is correlated with resistance with BMH-21, a novel anticancer drugs inhibiting Pol I, showing the relevance of targeting Pol I during transcriptional pauses to kill cancer cells. This work offers critical insights into Pol I dynamics, rRNA transcription regulation, and implications for cancer therapeutics.

We sincerely thank the three reviewers for their insightful comments and recognition of the strengths and weaknesses of our study. Their acknowledgment of our rigorous methodology, the relevance of our findings on rRNA transcription regulation, and the significant enzymatic properties of the SuperPol mutant is highly appreciated. We are particularly grateful for their appreciation of the potential scientific impact of this work. Additionally, we value the reviewer’s suggestion that this article could address a broad scientific community, including in transcription biology and cancer therapy research. These encouraging remarks motivate us to refine and expand upon our findings further.

All three reviewers acknowledged the increased processivity of SuperPol compared to its wildtype counterpart. However, two out of three questions our claims that premature termination of transcription can regulate ribosomal RNA transcription. This conclusion is based on SuperPol mutant increasing rRNA production. Proving that modulation of early transcription termination is used to regulate rRNA production under physiological conditions is beyond the scope of this study. Therefore, we propose to change the title of this manuscript to focus on what we have unambiguously demonstrated:

“Ribosomal RNA synthesis by RNA polymerase I is subjected to premature termination of transcription”.

Reviewer 1 main criticisms centers on the use of the CRAC technique in our study. While we address this point in detail below, we would like to emphasize that, although we agree with the reviewer’s comments regarding its application to Pol II studies, by limiting contamination with mature rRNA, CRAC remains the only suitable method for studying Pol I elongation over the entire transcription units. All other methods are massively contaminated with fragments of mature RNA which prevents any quantitative analysis of read distribution within rDNA. This perspective is widely accepted within the Pol I research community, as CRAC provides a robust approach to capturing transcriptional dynamics specific to Pol I activity.

We hope that these findings will resonate with the readership of your journal and contribute significantly to advancing discussions in transcription biology and related fields.

**Description of the planned revisions:**

Despite numerous text modification (see below), we agree that one major point of discussion is the consequence of increased processivity in SuperPol mutant on the “quality” of produced rRNA. Reviewer 3 suggested comparisons with other processive alleles, such as the *rpb1*-E1103G mutant of the RNAPII subunit (Malagon et al., 2006). This comparison has already been addressed by the Schneider lab (Viktorovskaya OV, *Cell Rep.*, 2013 - PMID: 23994471), which explored *Pol II* (*rpb1*-E1103G) and *Pol I* (*rpa190*-E1224G). The *rpa190*-E1224G mutant revealed enhanced pausing in vitro, highlighting key differences between Pol I and Pol II catalytic ratelimiting steps (see David Schneider's review on this topic for further details).

Reviewer 2 and 3 suggested that a decreased efficiency of cleavage upon backtracking might imply an increased error rate in SuperPol compared to the wild-type enzyme. Pol I mutant with decreased rRNA cleavage have been characterized previously, and resulted in increased errorrate. We already started to address this point. Preliminary results from in vitro experiments suggest that SuperPol mutants exhibit an elevated error rate during transcription. However, these findings remain preliminary and require further experimental validation to confirm their reproducibility and robustness. We propose to consolidate these data and incorporate into the manuscript to address this question comprehensively. This could provide valuable insights into the mechanistic differences between SuperPol and the wild-type enzyme. SuperPol is the first pol I mutant described with an increased processivity in vitro and in vivo, and we agree that this might be at the cost of a decreased fidelity.

Regulatory aspect of the process:

To address the reviewer’s remarks, we propose to test our model by performing experiments that would evaluate PTT levels in Pol I mutant’s or under different growth conditions. These experiments would provide crucial data to support our model, which suggests that PTT is a regulatory element of Pol I transcription. By demonstrating how PTT varies with environmental factors, we aim to strengthen the hypothesis that premature termination plays an important role in regulating Pol I activity.

We propose revising the title and conclusions of the manuscript. The updated version will better reflect the study's focus and temper claims regarding the regulatory aspects of termination events, while maintaining the value of our proposed model.

**Description of the revisions that have already been incorporated in the transferred manuscript:**

Some very important modifications have now been incorporated:

Statistical Analyses and CRAC Replicates:

Unlike reviewers 2 and 3, reviewer 1 suggests that we did not analyze the results statistically. In fact, the CRAC analyses were conducted in biological triplicate, ensuring robustness and reproducibility. The statistical analyses are presented in Figure 2C, which highlights significant findings supporting the fact WT Pol I and SuperPol distribution profiles are different. We CRAC replicates exhibit a high correlation and we confirmed significant effect in each region of interest (5’ETS, 18S.2, 25S.1 and 3’ ETS, Figure 1) to confirm consistency across experiments. We finally took care not to overinterpret the results, maintaining a rigorous and cautious approach in our analysis to ensure accurate conclusions.

CRAC vs. Net-seq:

Reviewer 1 ask to comment differences between CRAC and Net-seq. Both methods complement each other but serve different purposes depending on the biological question on the context of transcription analysis. Net-seq has originally been designed for Pol II analysis. It captures nascent RNAs but does not eliminate mature ribosomal RNAs (rRNAs), leading to high levels of contamination. While this is manageable for Pol II analysis (*in silico* elimination of reads corresponding to rRNAs), it poses a significant problem for Pol I due to the dominance of rRNAs (60% of total RNAs in yeast), which share sequences with nascent Pol I transcripts. As a result, large Net-seq peaks are observed at mature rRNA extremities (Clarke 2018, Jacobs 2022). This limits the interpretation of the results to the short lived pre-rRNA species. In contrast, CRAC has been specifically adapted by the laboratory of David Tollervey to map Pol I distribution while minimizing contamination from mature rRNAs (The CRAC protocol used exclusively recovers RNAs with 3′ hydroxyl groups that represent endogenous 3′ ends of nascent transcripts, thus removing RNAs with 3’-Phosphate, found in mature rRNAs). This makes CRAC more suitable for studying Pol I transcription, including polymerase pausing and distribution along rDNA, providing quantitative dataset for the entire rDNA gene.

CRAC vs. Other Methods:

Reviewer 1 suggests using GRO-seq or TT-seq, but the experiments in Figure 2 aim to assess the distribution profile of Pol I along the rDNA, which requires a method optimized for this specific purpose. While GRO-seq and TT-seq are excellent for measuring RNA synthesis and cotranscriptional processing, they rely on Sarkosyl treatment to permeabilize cellular and nuclear membranes. Sarkosyl is known to artificially induces polymerase pausing and inhibits RNase activities which are involved in the process. To avoid these artifacts, CRAC analysis is a direct and fully in vivo approach. In CRAC experiment, cells are grown exponentially in rich media and arrested via rapid cross-linking, providing precise and artifact-free data on Pol I activity and pausing.

Pol I ChIP Signal Comparison:

The ChIP experiments previously published in Darrière et al. lack the statistical depth and resolution offered by our CRAC analyses. The detailed results obtained through CRAC would have been impossible to detect using classical ChIP. The current study provides a more refined and precise understanding of Pol I distribution and dynamics, highlighting the advantages of CRAC over traditional methods in addressing these complex transcriptional processes.

BMH-21 Effects:

As highlighted by Reviewer 1, the effects of BMH-21 observed in our study differ slightly from those reported in earlier work (Ref Schneider 2022), likely due to variations in experimental conditions, such as methodologies (CRAC vs. Net-seq), as discussed earlier. We also identified variations in the response to BMH-21 treatment associated with differences in cell growth phases and/or cell density. These factors likely contribute to the observed discrepancies, offering a potential explanation for the variations between our findings and those reported in previous studies. In our approach, we prioritized reproducibility by carefully controlling BMH-21 experimental conditions to mitigate these factors. These variables can significantly influence results, potentially leading to subtle discrepancies. Nevertheless, the overall conclusions regarding BMH-21's effects on WT Pol I are largely consistent across studies, with differences primarily observed at the nucleotide resolution. This is a strength of our CRAC-based analysis, which provides precise insights into Pol I activity.

We will address these nuances in the revised manuscript to clarify how such differences may impact results and provide context for interpreting our findings in light of previous studies.

Minor points:
**Reviewer #1:**
In general, the writing style is not clear, and there are some word mistakes or poor descriptions of the results, for example:On page 14: "SuperPol accumulation is decreased (compared to Pol I)".On page 16: "Compared to WT Pol I, the cumulative distribution of SuperPol is indeed shifted on the right of the graph."

We clarified and increased the global writing style according to reviewer comment.

There are also issues with the literature, for example: Turowski et al, 2020a and Turowski et al, 2020b are the same article (preprint and peer-reviewed). Is there any reason to include both references? Please, double-check the references.

This was corrected in this version of the manuscript.

In the manuscript, 5S rRNA is mentioned as an internal control for TMA normalisation. Why are Figure 1C data normalised to 18S rRNA instead of 5S rRNA?

Data are effectively normalized relative to the 5S rRNA, but the value for the 18S rRNA is arbitrarily set to 100%.

Figure 4 should be a supplementary figure, and Figure 7D doesn't have a y-axis labelling.

The presence of all Pol I specific subunits (Rpa12, Rpa34 and Rpa49) is crucial for the enzymatic activity we performed. In the absence of these subunits (which can vary depending on the purification batch), Pol I pausing, cleavage and elongation are known to be affected. To strengthen our conclusion, we really wanted to show the subunit composition of the purified enzyme. This important control should be shown, but can indeed be shown in a supplementary figure if desired.

Y-axis is figure 7D is now correctly labelled

In Figure 7C, BMH-21 treatment causes the accumulation of ~140bp rRNA transcripts only in SuperPol-expressing cells that are Rrp6-sensitive (line 6 vs line 8), suggesting that BHM-21 treatment does affect SuperPol. Could the author comment on the interpretation of this result?

The 140 nt product is a degradation fragment resulting from trimming, which explains its lower accumulation in the absence of Rrp6. BMH21 significantly affects WT Pol I transcription but has also a mild effect on SuperPol transcription. As a result, the 140 nt product accumulates under these conditions.

**Reviewer #2:**
pp. 14-15: The authors note local differences in peak detection in the 5'-ETS among replicates, preventing a nucleotide-resolution analysis of pausing sites. Still, they report consistent global differences between wild-type and SuperPol CRAC signals in the 5'ETS (and other regions of the rDNA). These global differences are clear in the quantification shown in Figures 2B-C. A simpler statement might be less confusing, avoiding references to a "first and second set of replicates"

According to reviewer, statement has been simplified in this version of the manuscript.

Figures 2A and 2C: Based on these data and quantification, it appears that SuperPol signals in the body and 3' end of the rDNA unit are higher than those in the wild type. This finding supports the conclusion that reduced pausing (and termination) in the 5'ETS leads to an increased Pol I signal downstream. Since the average increase in the SuperPol signal is distributed over a larger region, this might also explain why even a relatively modest decrease in 5'ETS pausing results in higher rRNA production. This point merits discussion by the authors.

We agree that this is a very important discussion of our results. Transcription is a very dynamic process in which paused polymerase is easily detected using the CRAC assay. Elongated polymerases are distributed over a much larger gene body, and even a small amount of polymerase detected in the gene body can represent a very large rRNA synthesis. This point is of paramount importance and, as suggested by the reviewer, is now discussed in detail.

A decreased efficiency of cleavage upon backtracking might imply an increased error rate in SuperPol compared to the wild-type enzyme. Have the authors observed any evidence supporting this possibility?

Reviewer suggested that a decreased efficiency of cleavage upon backtracking might imply an increased error rate in SuperPol compared to the wild-type enzyme. We thank Reviewer #2 to point it as in our opinion, this is an important point what should be added to the manuscript. We have now included new data (panels 5G, 5H and 5I) in the manuscript showing that SuperPol in vitro exhibits an increased error rate compared to the WT enzyme. From these results obtained in vitro, we concluded that SuperPol shows reduced nascent transcript cleavage, associated with more efficient transcript elongation, but to the detriment of transcriptional fidelity.

pp. 15 and 22: Premature transcription termination as a regulator of gene expression is welldocumented in yeast, with significant contributions from the Corden, Brow, Libri, and Tollervey labs. These studies should be referenced along with relevant bacterial and mammalian research.

According to reviewer suggestion, we referenced these studies.

p. 23: "SuperPol and Rpa190-KR have a synergistic effect on BMH-21 resistance." A citation should be added for this statement.

This represents some unpublished data from our lab. KR and SuperPol are the only two known mutants resistant to BMH-21. We observed that resistance between both alleles is synergistic, with a much higher resistance to BMH-21 in the double mutant than in each single mutant (data not shown). Comparing their resistance mechanisms is a very important point that we could provide upon request. This was added to the statement.

p. 23: "The released of the premature transcript" - this phrase contains a typo

This is now corrected.

**Reviewer #3:**
Figure 1B: it would be opportune to separate the technique's schematic representation from the actual data. Concerning the data, would the authors consider adding an experiment with rrp6D cells? Some RNAs could be degraded even in such short period of time, as even stated by the authors, so maybe an exosome depleted background could provide a more complete picture. Could also the authors explain why the increase is only observed at the level of 18S and 25S? To further prove the robustness of the Pol I TMA method could be good to add already characterized mutations or other drugs to show that the technique can readily detect also well-known and expected changes.

The precise objective of this experiment is to avoid the use of the Rrp6 mutant. Under these conditions, we prevent the accumulation of transcripts that would result from a maturation defect. While it is possible to conduct the experiment with the Rrp6 mutant, it would be impossible to draw reliable conclusions due to this artificial accumulation of transcripts.

Figure 1C: the NTS1 probe signal is missing (it is referenced in Figure 1A but not listed in the Methods section or the oligo table). If this probe was unused, please correct Figure 1A accordingly.

We corrected Figure 1A.

Figure 2A: the RNAPI occupancy map by CRAC is hard to interpret. The red color (SuperPol) is stacked on top of the blue line, and we are not able to observe the signal of the WT for most of the position along the rDNA unit. It would be preferable to use some kind of opacity that allows to visualize both curves. Moreover, the analysis of the behavior of the polymerase is always restricted to the 5'ETS region in the rest of the manuscript. We are thus not able to observe whether termination events also occur in other regions of the rDNA unit. A Northern blot analysis displaying higher sizes would provide a more complete picture.

We addressed this point to make the figure more visually informative. In Northern Blot analysis, we use a TSS (Transcription Start Site) probe, which detects only transcripts containing the 5' extremity. Due to co-transcriptional processing, most of the rRNA undergoing transcription lacks its 5' extremity and is not detectable using this technique. We have the data, but it does not show any difference between Pol I and SuperPol. This information could be included in the supplementary data if asked.

"Importantly, despite some local variations, we could reproducibly observe an increased occupancy of WT Pol I in 5'-ETS compared to SuperPol (Figure 1C)." should be Figure 2C.

Thanks for pointing out this mistake. It has been corrected.

Figure 3D: most of the difference in the cumulative proportion of CRAC reads is observed in the region ~750 to 3000. In line with my previous point, I think it would be worth exploring also termination events beyond the 5'-ETS region.

We agree that such an analysis would have been interesting. However, with the exception of the pre-rRNA starting at the transcription start site (TSS) studied here, any cleaved rRNA at its 5' end could result from premature termination and/or abnormal processing events. Exploring the production of other abnormal rRNAs produced by premature termination is a project in itself, beyond this initial work aimed at demonstrating the existence of premature termination events in ribosomal RNA production.

Figure 4: should probably be provided as supplementary material.

As l mentioned earlier (see comments), the presence of all Pol I specific subunits (Rpa12, Rpa34 and Rpa49) is crucial for the enzymatic activity we performed. This important control should be shown, but can indeed be shown in a supplementary figure if desired.

"While the growth of cells expressing SuperPol appeared unaffected, the fitness of WT cells was severely reduced under the same conditions." I think the growth of cells expressing SuperPol is slightly affected.

We agree with this comment and we modified the text accordingly.

Figure 7D: the legend of the y-axis is missing as well as the title of the plot.

Legend of the y-axis and title of the plot are now present.

The statements concerning BMH-21, SuperPol and Rpa190-KR in the Discussion section should be removed, or data should be provided.

This was discussed previously. See comment above.

Some references are missing from the Bibliography, for example Merkl et al., 2020; Pilsl et al., 2016a, 2016b.

Bibliography is now fixed

**Description of analyses that authors prefer not to carry out:**

Does SuperPol mutant produces more functional rRNAs ?

As Reviewer 1 requested, we agree that this point requires clarification.. In cells expressing SuperPol, a higher steady state of (pre)-rRNAs is only observed in absence of degradation machinery suggesting that overproduced rRNAs are rapidly eliminated. We know that (pre)rRNas are unable to accumulate in absence of ribosomal proteins and/or Assembly Factors (AF). In consequence, overproducing rRNAs would not be sufficient to increase ribosome content. This specific point is further address in our lab but is beyond the scope of this article.

Is premature termination coupled with rRNA processing

We appreciate the reviewer’s insightful comments. The suggested experiments regarding the UTP-A complex's regulatory potential are valuable and ongoing in our lab, but they extend beyond the scope of this study and are not suitable for inclusion in the current manuscript.